# Strong chemotaxis by marine bacteria towards polysaccharides is enhanced by the abundant organosulfur compound DMSP

Estelle E. Clerc[1], Jean-Baptiste Raina [2] ✉, Johannes M. Keegstra [1], Zachary Landry[1], Sammy Pontrelli [3], Uria Alcolombri [1,4], Bennett S. Lambert[1], Valerio Anelli[1], Flora Vincent[5,6], Marta Masdeu-Navarro[7], Andreas Sichert [8], Frédéric De Schaetzen[1], Uwe Sauer[3], Rafel Simó [7], Jan-Hendrik Hehemann [8], Assaf Vardi [5], Justin R. Seymour [2] & Roman Stocker [1] ✉

The ability of marine bacteria to direct their movement in response to chemical gradients influences inter-species interactions, nutrient turnover, and ecosystem productivity. While many bacteria are chemotactic towards small metabolites, marine organic matter is predominantly composed of large molecules and polymers. Yet, the signalling role of these large molecules is largely unknown. Using in situ and laboratory-based chemotaxis assays, we show that marine bacteria are strongly attracted to the abundant algal polysaccharides laminarin and alginate. Unexpectedly, these polysaccharides elicited stronger chemoattraction than their oligo- and monosaccharide constituents. Furthermore, chemotaxis towards laminarin was strongly enhanced by dimethylsulfoniopropionate (DMSP), another ubiquitous algal-derived metabolite. Our results indicate that DMSP acts as a methyl donor for marine bacteria, increasing their gradient detection capacity and facilitating their access to polysaccharide patches. We demonstrate that marine bacteria are capable of strong chemotaxis towards large soluble polysaccharides and uncover a new ecological role for DMSP in enhancing this attraction. These navigation behaviours may contribute to the rapid turnover of polymers in the ocean, with important consequences for marine carbon cycling.

Chemical resources are often heterogeneously distributed at the microscale in the marine environment, which impacts microbial community composition and dynamics, and directly influences ecosystem productivity and biogeochemical cycles[1–3]. In this patchy seascape, some marine heterotrophic bacteria and archaea use chemotaxis, the ability to direct movement in response to chemical gradients, to exploit localised resources[4–8]. Small diffusible molecules, such as amino acids, simple sugars and secondary metabolites (e.g., the organosulfur compound dimethylsulfoniopropionate (DMSP)), have been identified as strong chemical cues for marine bacteria under laboratory conditions[9–13]. However, primary producers such as phytoplankton and algae also release large molecules and polymers[14],

[1]Institute of Environmental Engineering, Department of Civil, Environmental and Geomatic Engineering, ETH Zurich, Zurich, Switzerland. [2]Climate Change Cluster, University of Technology Sydney, Ultimo, Australia. [3]Institute of Molecular Systems Biology, Department of Biology, ETH Zurich, Zurich, Switzerland. [4]Institute for Life Sciences, Department of Plant and Environmental Sciences, The Hebrew University of Jerusalem, Jerusalem, Israel. [5]Department of Plant and Environmental Sciences, Weizmann Institute of Science, Rehovot, Israel. [6]Developmental Biology Unit, European Molecular Biological Laboratory, Heidelberg 69117, Germany. [7]Institut de Ciències del Mar, CSIC, Barcelona, Catalonia, Spain. [8]Max Planck Institute for Marine Microbiology, Bremen, Germany. ✉e-mail: Jean-Baptiste.Raina@uts.edu.au; romanstocker@ethz.ch

which represent a considerable fraction (25–37%) of the dissolved organic matter (DOM) in the ocean[15–18]. Yet, we do not know if bacteria can use chemotaxis to access this carbon pool.

Among large molecules, laminarin, a mannitol-containing β−1,3-linked polysaccharide consisting of 15–33 monomers of glucose[19,20], can account for up to 25% of the annual carbon pool derived from marine primary production[21]. As laminarin is a central storage compound for a diverse range of macro- and microalgae[22,23], it is often released in seawater upon cell lysis (e.g., during the collapse of phytoplankton blooms[21,24]) where it is rapidly catabolised by marine microorganisms[25–28]. Laminarin degradation by bacteria has been assumed to occur through random encounters, or following chemoattraction to smaller breakdown products[29,30]. However, limited laboratory-based reports of chemotaxis towards alginate and pectin[31], two other soluble polysaccharides, suggests that large polysaccharides, such as laminarin, may instead directly induce widespread chemical attraction among marine bacteria.

To determine whether polysaccharides attract bacteria in the ocean, we used the in situ chemotaxis assay[32,33] (ISCA), a microfluidic device consisting of an array of 25 micro-wells (110 μL each) that can be individually loaded with different chemicals[32]. Once the ISCA is deployed in an aqueous environment, the chemicals diffuse out of the wells through a narrow inlet, creating concentration gradients that can attract chemotactic microorganisms into the wells. We performed a systematic series of ISCA deployments during a phytoplankton bloom[34] induced in four mesocosm bags in the Norwegian fjord of Raunefjorden in the spring of 2018. At the start of the experiment, the natural fjord community within the mesocosms was enriched with nitrogen and phosphate, inducing a distinct succession in the phytoplankton community[34]: a first bloom dominated by the picophytoplankton *Bathycoccus* and *Micromonas* occurred rapidly, peaked on day 10, and was directly followed by a large bloom of the coccolithophore *Emiliania huxleyi* between days 10 and 22 (Fig. 1a).

To quantify the chemotactic response of bacteria to laminarin and to its constituent subunits, the disaccharide laminaribiose and the monomeric sugars glucose and mannitol, we deployed ISCAs in surface mesocosm water at seven different time points over 22 days during the course of the phytoplankton blooms (Fig. 1a), with each deployment lasting 2 h. Following laboratory-based and preliminary in situ tests (Fig. S1, Supplementary Data 1) and based on the molecular weight differences of the various substances used ("Methods" section), we selected a concentration of 10 mg mL$^{-1}$ for laminarin and 1 mg mL$^{-1}$ for laminaribiose, glucose and mannitol to reduce differences between substances in the amount of molecules loaded in the ISCA. Even with these concentrations, there were still only about half as many laminarin molecules as glucose or mannitol molecules in an ISCA well (Methods). Following deployment, the contents of the ISCA wells were recovered and aliquoted to (i) quantify the strength of chemotaxis towards each compound by enumerating microbial cells using flow cytometry; (ii) determine the composition of the bacterial communities using 16S rRNA gene amplicon sequencing; and (iii) isolate cells on selective media enriched with the tested compounds for follow-up laboratory experiments.

## Results and discussion
### In situ chemotaxis experiments
The polysaccharide laminarin induced a strong chemotactic response in natural microbial assemblages. The mean chemotactic index ($I_C$) for laminarin, defined as the ratio of the number of cells in the treatment wells to the number of cells in control wells containing filtered seawater (so that attraction corresponds to $I_C > 1$)[7,32,33], was $2.6 \pm 1.6$ (mean ± SD). Chemotaxis towards laminarin varied among deployments, with $I_C = 2.3 \pm 0.9$ on day 15 (just before the peak of the *E. huxleyi* bloom) and $5.3 \pm 1.5$ on day 22 (just after the peak; Fig. 1a, ANOVA, $p < 0.05$ for both days, Supplementary Data 2). The strength of

these chemotactic responses to laminarin are similar to that of marine bacteria to phytoplankton-derived dissolved organic matter[7] or rich culture medium[32] reported previously using the ISCA.

Of the four chemicals tested, laminarin was unexpectedly the only one that induced a positive chemotactic response (Fig. 1a, b, ANOVA, $p < 0.05$ for all comparisons with laminarin on days 15 and 22, Supplementary Data 2). In contrast, no significant chemoattraction was found towards the smaller oligomer laminaribiose or the monomers glucose and mannitol (Fig. 1b, ANOVA, $p > 0.05$, Supplementary Data 2). This finding contrasts with the strong chemoattraction towards small molecules often observed in laboratory settings[10–13]. Our in situ measurements comparing differently sized compounds sharing the same molecular building blocks (glucose and mannitol) thus provide direct evidence of a positive chemotactic response of marine bacteria to abundant polymers.

The bacterial communities attracted to laminarin were highly diverse. Laminarin-containing ISCA wells were enriched in copiotrophic bacteria, whereby the composition of the responding community was significantly different to the surrounding mesocosm seawater (Fig. 1c, on days 3, 9, 15, 18 and 22; PERMANOVA, $p < 0.001$, Supplementary Data 3) and from the seawater control wells (Fig. 1c, PERMANOVA, $p < 0.001$, Supplementary Data 4). We identified between 49 and 123 amplicon sequence variants (ASVs; based on 16S rRNA gene) enriched in the laminarin wells relative to the surrounding seawater at specific times during the bloom, as well as 67 ASVs enriched throughout the bloom (ANCOM-BC, $p < 0.05$, Supplementary Data 5–10). The 67 ASVs enriched in the laminarin treatments throughout the bloom belonged to 34 bacterial families and 13 classes. One third (11 of 34) belonged to the Gammaproteobacteria class and most others to Betaproteobacteria, Flavobacteria and Alphaproteobacteria. Some of the most strongly enriched ASVs belonged to the families Pseudomonadaceae (six ASVs, ranging from 6.5 to 10.2 log$_2$-fold enrichment), Alteromonadaceae (five ASVs, 0.3–5.2 log$_2$-fold enrichment), and Pseudoalteromonadaceae (two ASVs, 4.1–6.1 log$_2$-fold enrichment) (Fig. 1d, Supplementary Data 10). Members of these families are widely recognised for their motility and chemotactic performances[11,13,30,35]. Our results reveal that laminarin is a strong chemoattractant for a wide taxonomic range of marine bacteria.

### Chemotaxis and growth potential of laminarin size fractions in laboratory conditions
To confirm and further investigate the positive chemotactic response of marine bacteria towards polysaccharides, we carried out targeted chemotaxis experiments in the laboratory using four Gammaproteobacteria strains that we isolated from ISCA wells containing laminarin during our in situ experiments. The 16S rRNA gene sequences of these four strains fully matched (100% sequence identity, Supplementary Data 11) ASVs that were enriched in the laminarin wells compared to the surrounding seawater (ANCOM-BC, $p < 0.05$ for the four strains, Supplementary Data 5–6 and 8–9). These four strains included two *Pseudoalteromonas* (ASV16 and ASV39) and two *Alteromonas* (ASV76 and ASV109). Using the ISCA in laboratory experiments, we quantified the chemotactic response of these four strains to laminarin, laminaribiose and glucose (at the same concentrations used in situ), and additionally to 1 mg mL$^{-1}$ laminarin hexose, a six-subunit oligomer. In line with our in situ observations, laminarin induced the strongest chemotactic response among these chemicals (see Methods) for three out of the four isolates, with very high chemotactic indices: $I_C = 147.2 \pm 18.9$ for ASV16, $I_C = 208.7 \pm 56.9$ for ASV39, and $I_C = 15.5 \pm 4.3$ for ASV76 (Fig. 2a–c, ANOVA, $p < 0.05$ for all, Supplementary Data 12). Notably, these chemotactic indices to laminarin are the strongest ever reported across any compounds tested using the ISCA and are up to 11.5 times higher than the largest chemotactic response previously reported in laboratory conditions[33]. Among these

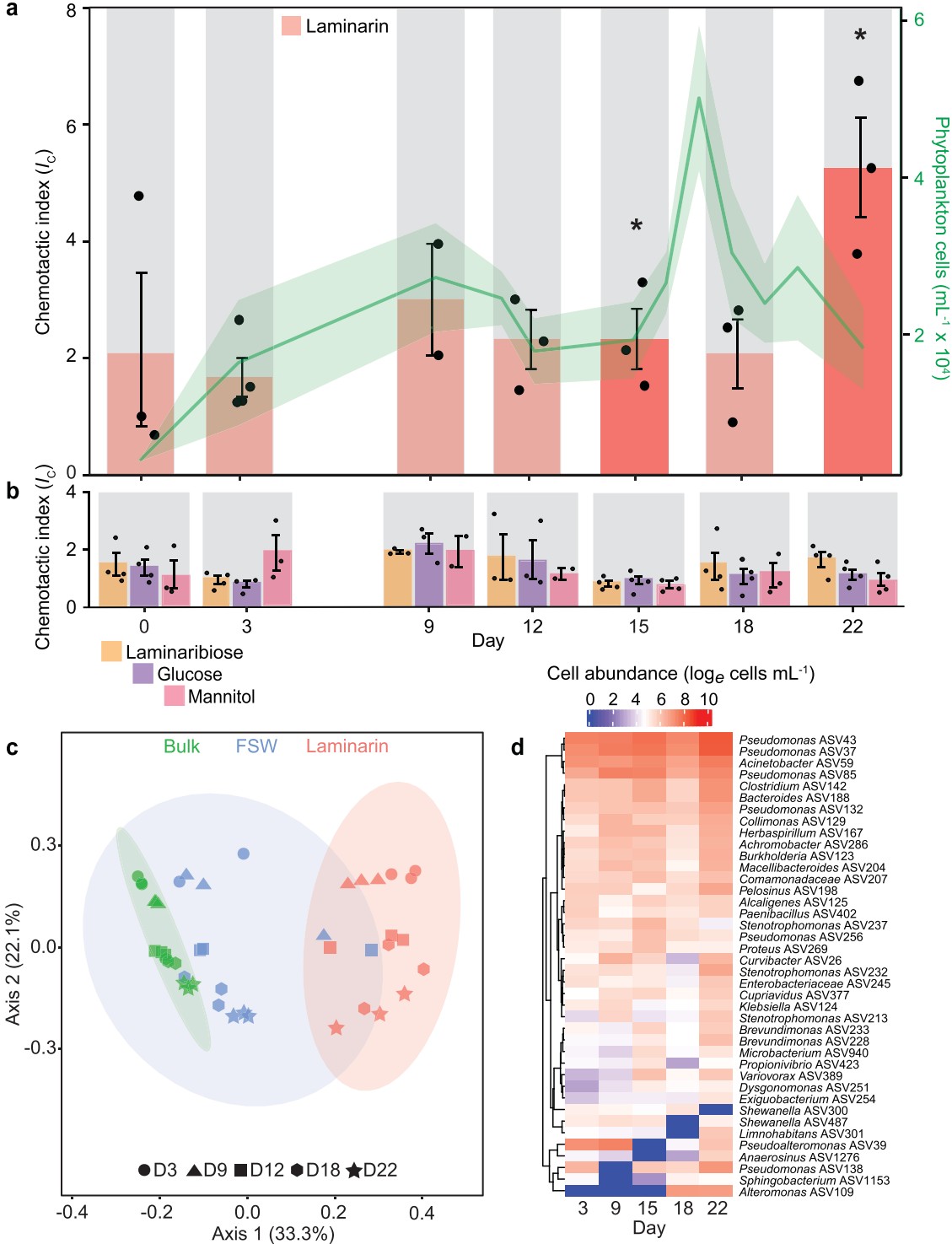

three strains, no significant chemotactic response was found to the hexose, the dimer or the monomer. In contrast, ASV109 responded to the hexose, but only modestly to laminarin ($I_C = 3.9 \pm 0.3$; Fig. 2d and Supplementary Data 12).

We next investigated whether the observed chemotactic preferences correspond to the ability of the different molecules to support growth, by culturing each of the four isolates with laminarin, laminaribiose and glucose as the sole carbon source (Fig. 2e–h). Significantly higher growth rates were observed for three out of four strains (ASV16, ASV76 and ASV109) after 48 h on laminarin compared to laminaribiose or glucose (all provided at 1 mg mL$^{-1}$, Fig. 2e, g, h, ANOVA, $p < 0.05$ for all comparisons with laminarin, Supplementary

Data 13). Over all four strains, growth rates on laminarin were on average 3.8 times higher than on laminaribiose and 3.0 times higher than on glucose (Fig. 2e–h and Supplementary Data 13). Yet importantly, growth on laminarin within the timescale of a laboratory ISCA experiment (1 h) was negligible for all four strains (Fig. S2, Supplementary Data 14), demonstrating that the number of cells in ISCA wells is a result of chemotaxis and not growth. These results demonstrate that laminarin not only elicited the strongest chemoattraction in three out of four strains, but also the strongest growth response compared to its building blocks for the culture conditions tested (Fig. 2e, g, h, ANOVA, $p < 0.05$ for all comparisons with laminarin, Supplementary Data 13).

**Fig. 1 | Laminarin induces the strongest chemotactic response in natural microbial communities throughout a phytoplankton bloom. a** Chemotactic index ($I_C$, red bars, left axis) measured in response to 10 mg mL$^{-1}$ laminarin throughout the phytoplankton bloom (green curve, right axis). The shaded green area depicts the standard deviation between triplicate biologically independent measurements ($n = 3$) of phytoplankton cell abundance at each time point. Days on which ISCAs were deployed are marked by grey shading. The chemotactic index $I_C$ denotes the average concentration of cells in the treatment wells of the ISCA, divided by the average concentration of cells in the control wells containing filtered seawater (FSW). Bars in darker shades denote a chemotactic index significantly larger than the FSW controls (ANOVA (one-sided), $p < 0.05$, all $p$-values are reported in Supplementary Data 2). **b** Chemotactic index for laminaribiose, glucose and mannitol over the course of the phytoplankton bloom. Each treatment was replicated across three different ISCAs ($n = 3$ biologically independent experiments,

individual dots). Data in panels a and b are mean ± SD. **c** Principal coordinates analysis (PCoA; Unifrac) showing that the bacterial communities responding to laminarin (red) are distinct from the surrounding seawater communities (green) (PERMANOVA, $p < 0.001$, all $p$-values are reported in Supplementary Data 3). The community in the wells containing filtered seawater (FSW, blue) represents cells that entered the ISCA wells by random motility (i.e., not driven by chemotaxis). Sampling days are indicated by different symbols. **d** Natural logarithm of absolute cell abundance of the 40 most strongly enriched bacterial ASVs (ANCOM-BC analysis, $p < 0.05$, all $p$-values are reported in Supplementary Data 10) in laminarin-containing ISCA wells compared to the surrounding bulk seawater communities, over the course of the phytoplankton bloom (natural log transformation, ASVs were clustered using Euclidian distance and ward.D2 clustering). Source data are provided as a Source Data file.

Laminarin is a mixture of polymers of different sizes[19,20] and may also contain smaller contaminants linked to the polymer's extraction process. To unambiguously confirm the chemoattractive properties of large laminarin molecules, we separated laminarin into three size fractions containing polymers (i) larger than ~ 18 glucose monomers (>3 kDa), (ii) smaller than ~18 monomers (<3 kDa) and (iii) between ~ 6 and 18 monomers (1–3 kDa). We then tested the chemotactic response of the four bacterial isolates to each fraction using the ISCA. Both the >3 kDa and <3 kDa fractions elicited a significant chemotactic response in comparison with the artificial seawater control (Fig. 2a–c, ANOVA, $p < 0.05$ for ASV16, ASV39 and ASV76, Supplementary Data 12). However, the largest size fraction (>3 kDa) induced the strongest chemotactic response for ASV16, ASV39 and ASV76 (Fig. 2a–c, ANOVA, $p < 0.05$, Supplementary Data 12), with the latter two strains displaying a response comparable in magnitude to unfractionated laminarin (with $I_C$ values as high as 172.4 ± 2.0; Fig. 2b, c, ANOVA, $p > 0.05$ for ASV39 and ASV76, Supplementary Data 12). These results confirm that the largest laminarin oligomers were responsible for the positive chemotactic response we observed in situ (Fig. 1a, Supplementary Data 2).

Preferential chemotaxis to the larger size fractions of laminarin, and to laminarin overall compared to its component monomers and oligomers, is surprising, as small, highly diffusive compounds have been traditionally considered most effective signals and growth substrates in the marine environment due to their passive diffusion through membrane porins[36], which allows quick assimilation by the cells[37]. In contrast, polysaccharides require first active transport into the periplasm by transmembrane proteins in order to be sensed by the cells[28,31,38,39]. The strong response of three of our isolates (ASV16, ASV39 and ASV76) to the largest polymer size could therefore indicate whole polymer uptake and sensing, possibly mediated through the SusD-binding protein[39].

Further laboratory-based experiments with the four isolates revealed that preferential chemotaxis to polymers over oligomers is not restricted to laminarin, but also occurs for the highly abundant marine polysaccharide alginate[40] (Supplementary Note 1, Fig. S3, ANOVA, $p < 0.05$, Supplementary Data 15), suggesting that this may be a common phenomenon in the ocean. These results expand the array of ecologically-relevant chemical cues beyond the small metabolites that have traditionally been investigated, showing that highly prevalent polymers attract more chemotactic cells than their highly diffusive building blocks. This stronger attraction might be related to polymers' lower diffusivity, which results in steeper and longer-lasting gradients[1,5,41].

### Influence of dimethylsulfoniopropionate on chemotactic responses

The magnitude of bacterial chemotaxis to laminarin was variable during the phytoplankton bloom (Fig. 1a). To determine if environmental factors influenced the strength of the chemotactic response to laminarin, we performed a correlative analysis with 61 biological (e.g.,

bacterial, viral and phytoplankton concentrations), physical (e.g., temperature, salinity, density, turbidity) and chemical (e.g., concentrations of particulate and dissolved organic carbon, total nitrogen, glycans) parameters recorded daily throughout the campaign[34]. The concentrations of glucose and 13 other glycans correlated negatively with the chemotactic index to laminarin (Spearman's correlations (two-sided), $R^2 = -0.93$ to $-0.60$, all $p < 0.01$; Fig. 3a, Supplementary Data 16). When abundant in seawater, these molecules may constitute signals that compete with laminarin gradients, such as those arising from ISCA wells. Conversely, the concentration of dissolved dimethylsulfoniopropionate ($DMSP_d$) showed the strongest positive correlation with the chemotactic index to laminarin (Spearman's correlation (two-sided), $R^2 = 0.90$, $p < 0.05$; Figs. 3a, S4, Supplementary Data 16), followed by total and particulate DMSP concentrations (Fig. 3a, Spearman's correlations (two-sided), $R^2 = 0.58$ for both, $p < 0.01$, Supplementary Data 16). This observation was intriguing given that DMSP is itself a potent behavioural cue[13,42]. This molecule is widely produced by phytoplankton, is one of the most abundant reduced sulfur compounds in the ocean[43,44], and an important nutrient source for marine microorganisms[45]. In addition, DMSP has often been reported for its chemoattractive properties, not only for bacteria, but also for marine protists[13,42,46] and even fishes[47]. The multifaceted ecological importance of DMSP has been described in several reviews[48,49].

During the field experiments, $DMSP_d$ concentrations peaked on days 8, 16 and 22, which coincided with the three blooms of photosynthetic microorganisms and with the enrichment of the four isolates in ISCA wells containing laminarin (average log$_2$ fold change relative to the surrounding seawater), pointing towards a potential dependence in the abundance of these strains on $DMSP_d$ concentration (Fig. 3b and Supplementary Data 7–8 and 17). To elucidate the role played by $DMSP_d$ in the chemotaxis of bacteria to laminarin, we carried out further laboratory ISCA experiments with the four bacterial isolates (Fig. 3c–f). We quantified their chemotactic response towards laminarin (10 mg mL$^{-1}$) when the background artificial seawater (outside of the ISCA wells, Supplementary Note 2) was spiked with $DMSP_d$ at different concentrations (0.1, 1 and 10 μM), representative of phytoplankton lysis events[50], yet not sufficient to induce bacterial growth within 1 h (Fig. S5 and Supplementary Data 18). Remarkably, the presence of $DMSP_d$ resulted in a substantial enhancement in the levels of the chemotaxis to laminarin for all strains (Fig. 3c–f, Supplementary Note 3, ANOVA, $p < 0.05$ for all strains, Supplementary Data 19). For example, the response of ASV16 to laminarin was 2.5 times stronger in the presence of 1 μM $DMSP_d$ compared to the case in which $DMSP_d$ was not added ($I_C = 147.3 ± 31.7$ with 1 μM $DMSP_d$ vs. $I_C = 57.9 ± 6.3$ without $DMSP_d$; Fig. 3c, Supplementary Data 19). Furthermore, a 1.8-fold, $DMSP_d$-mediated enhancement of chemotaxis was found on average also for chemotaxis to 1 mg mL$^{-1}$ alginate ($I_C = 21.7 ± 9.7$ with 10 μM $DMSP_d$ vs. $I_C = 11.9 ± 3.9$ without $DMSP_d$; Fig. S3a–d, ANOVA, $p < 0.05$, Supplementary Data 20). In contrast, we generally observed no increase in the strength of chemotaxis towards the monomeric units of

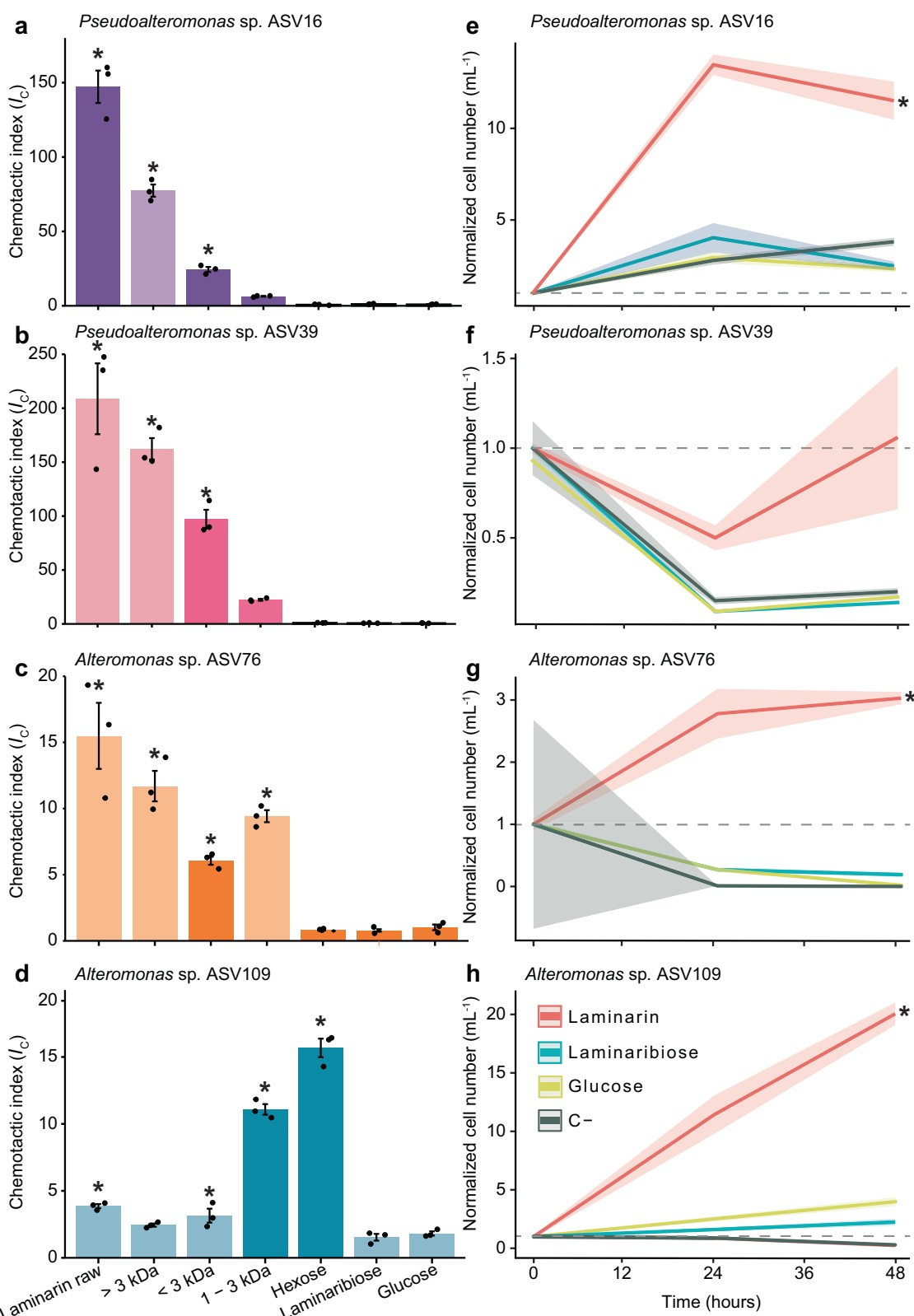

these polymers (glucose and mannuronate, respectively) following addition of $DMSP_d$ at any of the concentrations tested (Figs. S3i–l and S6i–l, Supplementary Data 19 and 21). The only two exceptions occurred for ASV16 and ASV76 that chemotaxed to glucose when 10 μM $DMSP_d$ were present (10.6-fold and 1.64-fold enhancement, respectively) (Fig. S6i, k and Supplementary Data 19). These results reveal a synergistic effect of $DMSP_d$ on chemotaxis to soluble

polysaccharides, whereby the presence of $DMSP_d$ enhances the ability of marine bacteria to migrate towards higher polymer concentrations (Supplementary Note 4).

**How does DMSP affect chemotaxis to soluble polysaccharides?**
In pursuing the mechanism underlying this synergistic effect, we first hypothesised that the addition of $DMSP_d$ may increase the bacterial

**Fig. 2 | The larger laminarin polymers elicit stronger chemotactic responses and growth rates. a–d** Chemotactic index ($I_C$, coloured bars, left axis) in laboratory ISCA experiments in response to different laminarin size fractions (>3 kDa, <3 kDa, 1–3 kDa), raw laminarin (i.e., unfractionated), laminarin hexose, laminaribiose, and glucose, for *Pseudoalteromonas* sp. ASV16 (**a**), *Pseudoalteromonas* sp. ASV39 (**b**), *Alteromonas* sp. ASV76 (**c**) and *Alteromonas* sp. ASV109 (**d**). Bars marked with an asterisk denote $I_C$ values significantly greater than the artificial seawater controls (ANOVA (one-sided), $p < 0.05$, all $p$-values are reported in Supplementary Data 12). Bars in darker shades denote $I_C$ values significantly different from the >3 kDa polymer fraction (ANOVA, $p < 0.05$, all $p$-values are reported in Supplementary Data 12). Each treatment was replicated across three different ISCAs ($n = 3$

biologically independent experiments, individual dots). Data are mean ± SD. **e–h** Growth curves for ASV16 (**e**), ASV39 (**f**), ASV76 (**g**) and ASV109 (**h**) on raw (unfractionated) laminarin and its constituents laminaribiose and glucose (all three provided at 1 mg mL$^{-1}$), measured by flow cytometry over 48 h. For each curve, cell counts (cells mL$^{-1}$) were normalised by dividing by the count at time zero. Data are mean of triplicate measurements ± SD in shaded area. Growth rates of ASV16, ASV76 and ASV109 between 0 and 48 h in laminarin were significantly higher than in the other treatments (glucose, laminaribiose and the carbon-free control) as denoted by an asterisk (ANOVA (one-sided), all $p < 0.01$, all $p$-values are reported in Supplementary Data 13). Source data are provided as a Source Data file.

swimming speed, which has been shown to enhance the chemotactic precision for some bacteria[42]. However, cell tracking of ASV39 and ASV16 revealed no increase in either the swimming speed, the proportion of motile cells or their reorientation frequency (Figs. 4a and S7) compared to DMSP$_d$-free controls. A second hypothesis came from classic studies on *E. coli*[10,51,52], which showed that addition of methionine enhances chemotaxis of *E. coli* to other chemicals. These studies revealed that the sensitivity of chemoreceptors (methyl-accepting chemotaxis proteins, MCPs) in *E. coli* strongly depends on the availability of methyl groups provided by S-adenosyl-methionine (SAM), a methyl donor that is derived from methionine[53], which occurs in high intracellular concentration in bacteria and is involved in many cellular processes[53–55]. Methylation of MCPs allows the cell to adapt its sensing ability as it climbs a chemical gradient[56,57], promoting chemotaxis. We thus hypothesised that DMSP serves a similar purpose through the provision of methyl groups that ultimately enhance the chemotaxis of the bacteria. Using ISCA experiments with *E. coli*, we were first able to reproduce the dependence of its chemotaxis towards 1 mM aspartate on the concentration of methionine (Fig. S8a, ANOVA, $p < 0.001$, Supplementary Data 22). Then, in further experiments we found that replacing methionine with DMSP$_d$ (which contains two methyl groups) also enhanced the levels of chemotaxis by *E. coli* towards 1 mM aspartate (Fig. S8b, ANOVA, $p < 0.001$, Supplementary Data 22) to a similar level as the influence of methionine[58] (Fig. S8, ANOVA, $p > 0.05$ for 1 mM DMSP$_d$ and 1 mM methionine, Supplementary Data 22). Finally, we measured the strength of the chemotactic response to laminarin for the four marine isolates upon an addition of methionine (0.1, 1 and 10 µM), rather than DMSP$_d$. We found that the chemotactic index to laminarin was 1.6 to 4.0 times higher for three of the four bacteria tested (ASV16, ASV39 and ASV76) upon addition of 1 µM methionine (Figs. 4b and S9a, c, ANOVA, $p < 0.05$ for the three strains, Supplementary Data 23). Only ASV109 did not show an increase in chemotaxis, in line with its weaker chemotactic response to laminarin (Fig. S9b and Supplementary Data 23). These results support the hypothesis that DMSP$_d$ enhances the chemotaxis of marine bacteria to laminarin by acting as a source of methyl groups that favour sensory adaptation.

To further corroborate this hypothesis, we tested the effect of different concentrations of choline, a nitrogenated amino acid with three methyl groups, and homocysteine, a homologue of methionine lacking methyl groups, on the chemotactic response of ASV16 to laminarin. Transformation of choline into methionine is mediated via the glycine betaine pathway[59] and is one of many mechanisms making methionine available to microbial communities[60]. Choline had a similar effect as DMSP$_d$ and methionine, whereby it increased chemotaxis to laminarin by a factor of 2.1 ± 0.2 and 3.1 ± 0.3 when present at a concentration of 1 and 10 µM, respectively (Fig. 4c, ANOVA, $p < 0.05$ for both, Supplementary Data 24). In contrast, homocysteine did not lead to an increase in chemotaxis to laminarin (Fig. 4d, ANOVA, $p > 0.05$, Supplementary Data 25). Finally, using an in vitro enzyme assay employing *Pseudoalteromonas* sp. ASV16 cell lysate as an enzymatic source, we demonstrated that the cell lysate is capable of producing

methionine only when supplied with DMSP and homocysteine in combination (Fig. S10d). These results provide direct evidence that ASV16 possesses enzymes able to transfer a methyl group from DMSP to homocysteine to form methionine. This reaction is likely mediated through a yet uncharacterised pathway, as all close relatives of the marine strains used here lack the *dmdA* gene, the only gene known to demethylate DMSP[48] (Supplementary Data 26).

These findings strongly support the hypothesis that DMSP enhanced chemotaxis towards laminarin by supplying methyl groups to MCPs. We propose that in the ocean, DMSP$_d$ (and potentially other methylated compounds) can be used by marine bacteria as a precursor for methylation of the chemotactic machinery, enhancing their response to polymer gradients (Fig. 4e). This effect can be significant, with, for example, chemotaxis to laminarin 4.1-fold stronger in ASV39 in the presence of 0.1 µM DMSP$_d$ and 3.1-fold stronger in ASV76 in the presence of 10 µM DMSP$_d$ (Fig. 3d, e and Supplementary Data 19). Furthermore, the advantage provided by DMSP$_d$ is expected to be stronger for steep chemical gradients, for which the chemotactic machinery requires more rapid adaptation[57]. This is significant because slowly diffusing polymers such as laminarin and alginate will form sharper gradients than monomers and small oligomers[41]. In-line with this, our experiments reveal that the chemotactic advantage provided by DMSP$_d$ is indeed stronger for polymers than for monomers.

In summary, our results indicate that bacterial chemotaxis to soluble polysaccharides may be surprisingly common in the ocean and reveal that marine heterotrophic bacteria are able to use these abundant compounds as both chemotactic cues and growth substrates, favouring sustained polymer degradation and population growth. Furthermore, we demonstrate that chemotaxis to soluble polysaccharides is considerably enhanced by the presence of DMSP$_d$ and provide evidence that this enhancement occurs through the supply of methyl groups to chemotactic cells. As DMSP is ubiquitous in seawater[45], we expect DMSP-enhanced chemotaxis to be widespread in the ocean, and to apply also to other bacteria and other chemicals beyond laminarin and alginate. This novel role of DMSP provides, to the best of our knowledge, the first example of multiple phytoplankton metabolites synergistically affecting bacterial behaviour. Bacterial chemotaxis and its modulation by the methylation of the chemotaxis machinery is likely to play an important role in the turnover of abundant polysaccharides, with major influence on the oceanic carbon cycle.

## Methods
### ISCA fabrication
VeroGray polymer was used to create 3D-printed moulds on an Objet30 3D printer (Stratasys), using previously described protocols[32]. Each ISCA consisted of 25 wells arranged in a 5 × 5 array. Each 110 µL well possessed a 800-µm-diameter port that connected the inside of the well with the surrounding seawater and allowed the chemicals loaded in the wells to diffuse and create chemical gradients mimicking those present around marine particles and phytoplankton cells[5,32]. The

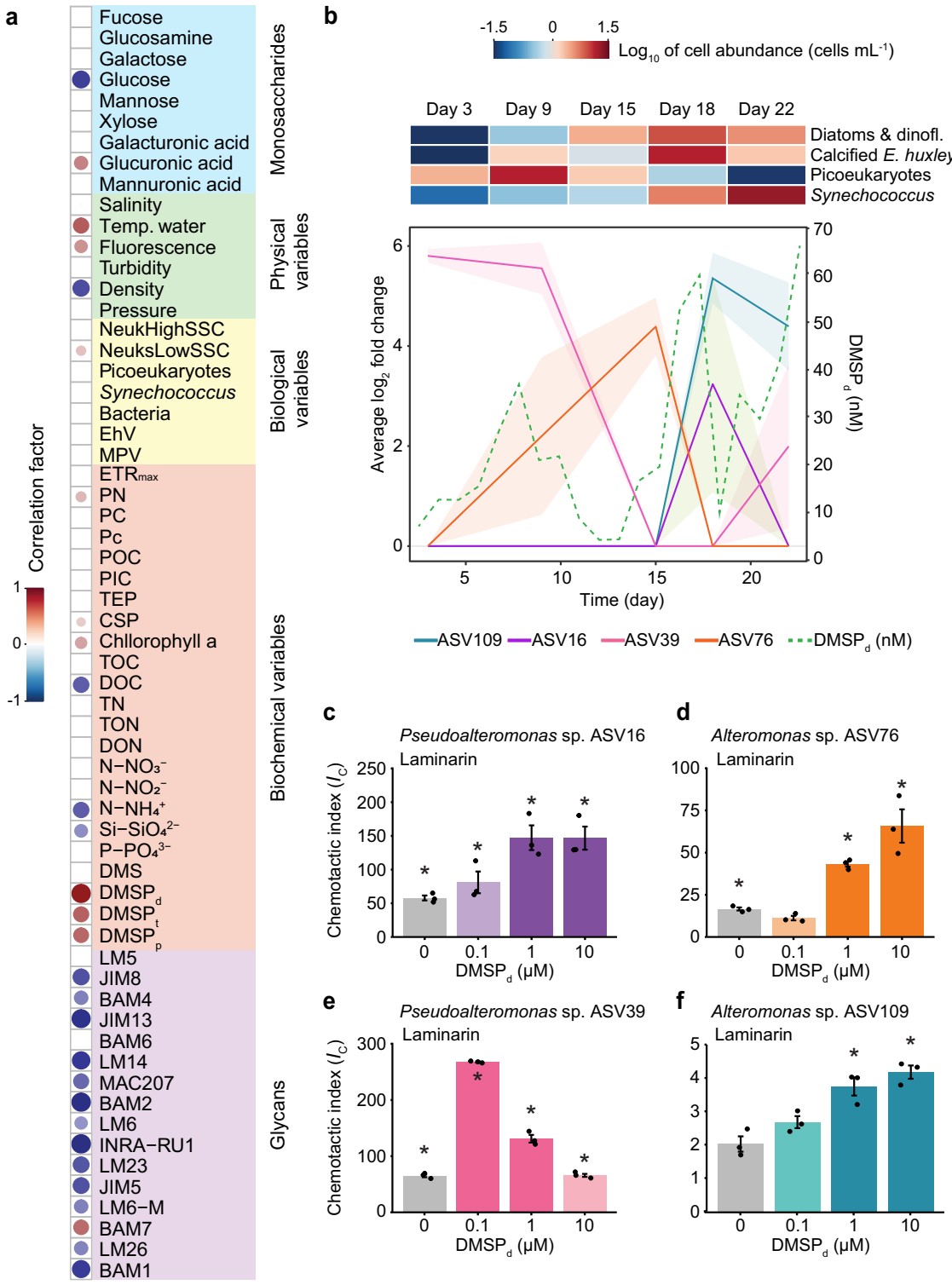

polydimethylsiloxane (PDMS) ISCA was fabricated by pouring 25 g PDMS (10:1 base to curing agent, wt/wt; Sylgard 184, Dow Corning) onto the mould and curing at 40 °C overnight. The cured PDMS slab (95 mm × 65 mm × 4.6 mm) was removed from the mould using a razor blade and the ports were cleared if needed using a biopsy punch (ProSciTech). The PDMS part was then UV-sterilised and bonded to a glass slide (100 mm × 76 mm × 1 mm, VWR) using a plasma oven (5 min, Harrick Scientific). The ISCAs were then incubated on a heating plate at 90 °C for 1 h to ensure full bonding before being stored with a protective layer of adhesive tape (Scotch) at room temperature until use.

**General ISCA deployment procedure**

Before use in experiments, chemoattractants were filtered with a 0.2 µm syringe (Millipore) to remove particles and potential contaminants. Within the 5 × 5 array of ISCA wells, one full row of five wells was allocated to each compound as technical replicates. For experiments conducted in laboratory conditions, the chemoattractants were injected in triplicate ISCAs with a sterile 1 mL syringe (Codau) and needle (27G, Henke Sass Wolf). Experiments were conducted by incubating the ISCAs for 1 h in the bacterial cultures, which had previously been diluted in artificial seawater (Instant Ocean, Spectrum

**Fig. 3 | Dissolved DMSP (DMSP$_d$) significantly enhances the strength of chemotaxis to laminarin. a** Correlation of 61 environmental variables recorded during the bloom with the chemotactic index elicited by laminarin. All variable names (abbreviations) are fully presented in Supplementary Data 16. Dot size and colour shading indicate the correlation strength, with positive correlations in red or negative ones in blue. Only significant correlations are displayed (Spearman's correlation (two-sided), $p < 0.05$, Supplementary Data 16). The environmental concentrations of DMSP$_d$ (dissolved), DMSP$_t$ (total) and DMSP$_p$ (particulate) exhibited a strong positive correlation with the strength of chemotaxis to laminarin (Spearman's correlation (two-sided), all $p < 0.01$, Supplementary Data 16). **b** Microbial cell abundance through the campaign. Average $\log_2$-fold change (relative to surrounding seawater) of the concentration of four ASVs (left axis) isolated from laminarin wells, throughout the bloom: *Pseudoalteromonas* sp. ASV16 (purple), *Pseudoalteromonas* sp. ASV39 (pink), *Alteromonas* sp. ASV76 (orange), and *Alteromonas* sp. ASV109 (blue). Shading indicates the standard deviation

between triplicate measurements at each time point. The green dashed line is the DMSP$_d$ concentration (right axis). The shift in phytoplankton taxonomy and abundance is shown on the heat map. **c–f** Chemotactic index ($I_C$, coloured bars, left axis) in response to 10 mg mL$^{-1}$ laminarin in laboratory experiments in the presence of different concentrations of DMSP$_d$ (0, 0.1, 1 and 10 μM) in the surrounding artificial seawater, for *Pseudoalteromonas* sp. ASV16 (**c**), *Alteromonas* sp. ASV76 (**d**), *Pseudoalteromonas* sp. ASV39 (**e**) and *Alteromonas* sp. ASV109 (**f**). An asterisk denotes a chemotactic index significantly larger than the artificial seawater controls (ANOVA, $p < 0.05$, Supplementary Data 19). Bars in darker shades indicate that the chemotactic response to laminarin was significantly larger than the treatment without DMSP$_d$ (ANOVA (one-sided), $p < 0.05$, all $p$-values are reported in Supplementary Data 19). Each treatment was replicated across three different ISCAs ($n = 3$ biologically independent experiments, denoted by individual dots). Bar plots represent mean ± SD. Source data are provided as a Source Data file.

Brands) at a concentration of $10^6$ cells mL$^{-1}$. After this incubation time, the contents of the ISCA wells were retrieved with a sterile syringe and needle and transferred to 1 mL Eppendorf tubes, by pooling a row of 5 wells containing the same attractants. The samples were stained with SYBR Green I (Thermofisher) and the chemotactic response was quantified by counting cells using flow cytometry.

For deployments conducted in situ, four ISCAs were deployed simultaneously to act as biological replicates. The ISCAs were placed into a flow-damping acrylic enclosure, following our pre-established procedure[32,33]. Each enclosure was filled with water coming from the four different mesocosm bags, previously mixed in equal volume. The flow-damping enclosures were filled slowly with water from the inlet in the lower surface, and were sealed with a plug once full. The enclosures were then incubated in situ for 2 h at 1 m depth. Following deployments, enclosures were retrieved and slowly drained. The volume of each ISCA well was retrieved using a sterile 1 mL syringe and 27 G needle. The contents of five wells per ISCA, containing the same treatment, were pooled (total 550 μL) and divided immediately into: (i) a 130 μL aliquot for flow cytometry, fixed with glutaraldehyde (2% final concentration) and snap frozen in liquid nitrogen; (ii) a 300 μL aliquot for DNA extraction and 16S rRNA gene sequencing, snap frozen immediately; and (iii) a 120 μL aliquot used for isolation of chemotactic strains. Additionally, bulk samples of 500 μL from the mixed mesocosm water were collected at each deployment for flow cytometry and DNA sequencing. Temperature, salinity and other water parameters were recorded with a CTD probe[34].

## ISCA preliminary experiments

To determine the optimal concentrations of the different compounds for chemotaxis assays in situ, we performed preliminary experiments under laboratory conditions using the highly motile marine isolate *Vibrio coralliilyticus* (YB2). *V. coralliilyticus* YB2 has been used extensively in experiments with the ISCA[32] and belongs to the family Vibrionaceae, which was highly enriched in ISCA wells containing phytoplankton-derived dissolved organic matter in previous in situ chemotaxis experiments[7]. The strain was inoculated on Marine Agar plates (BD Difco) from a glycerol stock and incubated for 16 h at 30 °C. A single colony was then transferred to 10% Marine Broth (BD Difco) in 0.22-μm-filtered artificial seawater (Instant Ocean, Spectrum Brands) and grown overnight at 30 °C in a shaking incubator. The culture was diluted 1/1000 (vol/vol) in 0.22-μm-filtered artificial seawater to obtain the bacterial suspension used in the experiments.

A concentration range was first tested under laboratory conditions for laminarin, glucose and mannitol. Each compound was resuspended at a concentration of 10 mg mL$^{-1}$ in sterile-filtered artificial seawater (Instant Ocean, Spectrum Brands) and filtered with a 0.2 μm Millex FG filter (Millipore) to remove particles. The filtrates were then serially diluted from 10 mg mL$^{-1}$ to 10 μg mL$^{-1}$ (final concentrations). Each concentration was loaded into three different ISCAs using

single-use 27 G needles (Henke Sass Wolf) connected to a 1 mL syringe (Codau). The ISCAs were incubated individually for an hour in the diluted *V. coralliilyticus* culture (see above) in 200 mL-capacity trays. The contents of the wells were retrieved and fixed with filtered glutaraldehyde (2% final concentration), and then cell concentrations in the wells were quantified using flow cytometry (Fig. S1a–c and Supplementary Data 1).

These concentration tests were carried out one more time at the field site before the start of the experiment. We repeated a concentration range with the three substances, as well as laminaribiose, to confirm which concentration elicited the strongest chemotactic responses (Fig. S1d–g).

We also performed growth tests with the fjord bacterial community and the predetermined concentrations of each substance, to confirm that no growth occurred on the timescale of the in situ incubation period (2 h). During these tests, we incubated fjord water samples with the 10% Difco 2216 Marine Broth (BD Diagnostics) within ISCA in situ for 2 h to mimic the experimental conditions. Triplicate samples were taken before incubation, and after 2 h. Samples were then fixed with filtered glutaraldehyde (2% final concentration) and cells counted using flow cytometry.

## ISCA field experiments

Field deployments were conducted over 22 days in May–June 2018 at the Espegrend Marine Research Field Station (60°16'10.3"N; 5°13'22.4"E) near Bergen, Norway. Non-permeable uncovered seawater mesocosms (11 m$^3$, 4 m deep and 2 m wide, 90% photosynthetically active radiation) were filled with seawater taken from the immediate surroundings in the fjord coastal environment. A phytoplankton bloom was induced artificially by adding phosphate and nitrate until day 7 and 5, respectively, and again on days 14–17[34]. The bags were sampled every day for 24 days, covering a set of 61 biogeochemical parameters[34], including the quantification of bacteria and phytoplankton by flow cytometry, and community composition via 16S and 18S rRNA gene amplicon sequencing.

ISCAs were deployed on days 0, 3, 9, 12, 15, 18 and 22, which covered the full phytoplankton bloom. For each deployment, mesocosm water from the four bags was sampled, mixed in equal volumes and subjected to a quadruple-filtration process, where 50 mL was first filtered through a 0.2 μm Millex FG (Millipore) to remove large particles, followed twice by a 0.2 μm Sterivex filter (Millipore) and then through a 0.02 μm Anatop filter (Whatman). This quadruple filtration aimed to remove all microorganisms from the seawater. The filtered seawater (FSW) was used as negative control in the ISCA, and was also used to resuspend the chemoattractants at their predetermined concentrations.

Because we tested compounds of different sizes (glucose: 180.1, mannitol: 182.2, laminaribiose: 342.3, laminarin: 2900–6200 g mol$^{-1}$), it was not possible to normalise the different compounds by weight,

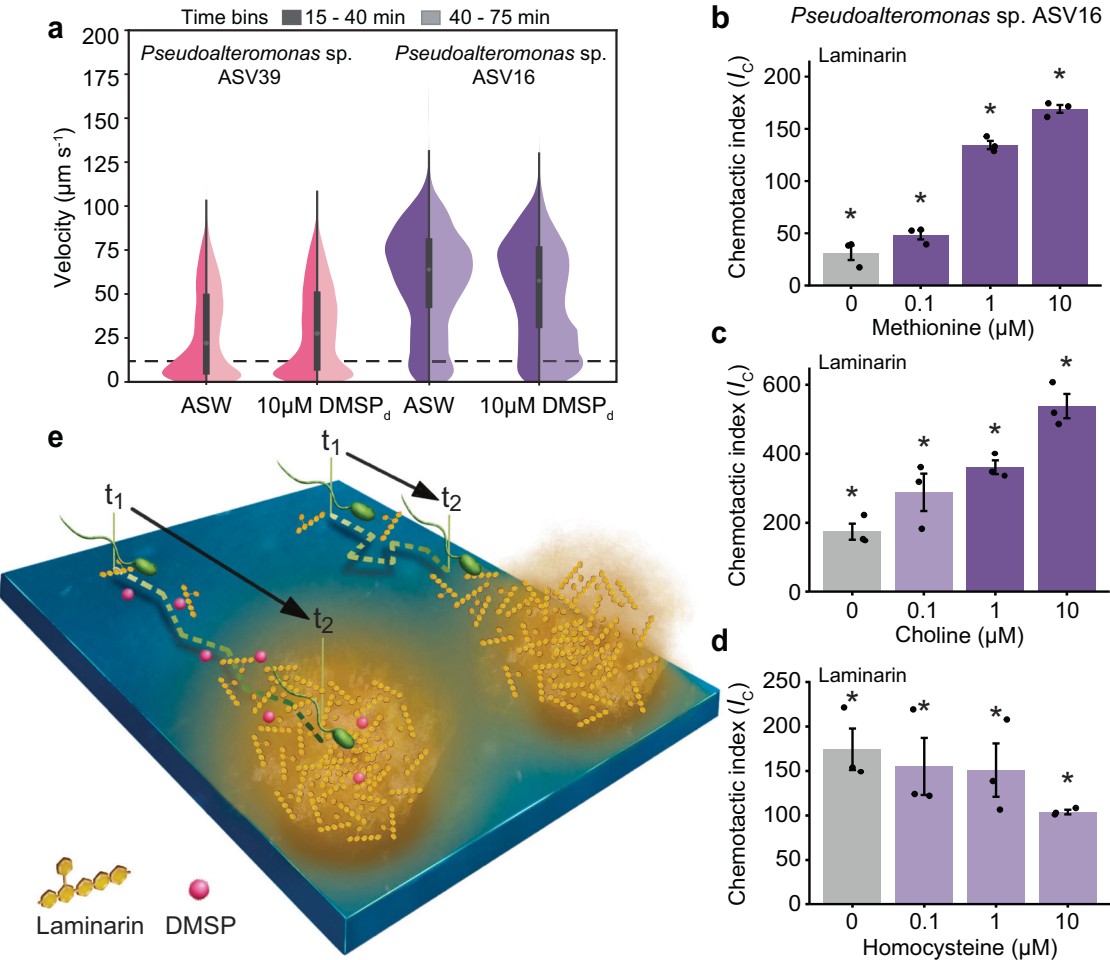

**Fig. 4 | Chemotaxis to laminarin is enhanced by other methylated compounds.** **a** Distribution of swimming velocity (left axis) of two environmental isolates (*Pseudoalteromonas* sp. ASV39 and ASV16) in the presence and absence of 10 µM DMSP$_d$. Overnight cell cultures were diluted to $10^6$ cells mL$^{-1}$ and amended with 10 µM (final concentration) of DMSP$_d$ ($n_{total}$ = 2.3 × $10^4$ and 2.4 × $10^3$ cells for *Pseudoalteromonas* sp. ASV39 and ASV16, respectively). A second culture in artificial seawater (ASW) served as control ($n_{total}$ = 1.6 × $10^4$ and 4.2 × $10^3$ cells for *Pseudoalteromonas* sp. ASV39 and ASV16, respectively). Cells from each culture were tracked by video-microscopy, between 15 and 40 min after DMSP$_d$ amendment (dark shade), and between 40 and 75 min (light shade). Box plots inside the violins indicate the quartiles of corresponding measurements. Minimum and maximum values are indicated by endpoints; centre dot denotes median values; whiskers denote 1.5 × the interquartile range; fine line extends to outlier measurements. Results show no significant difference in swimming velocity between the DMSP$_d$-amended treatment and the DMSP$_d$-free control (Fig. S7, ANOVA (one-sided), $p$ > 0.05). **b**−**d** Chemotactic index ($I_C$, coloured bars, left axis) in response to 10 mg mL$^{-1}$ laminarin in laboratory experiments in the presence of different concentrations (0, 0.1, 1 and 10 µM) of (**b**) methionine (one methyl group), (**c**) choline (three methyl groups), and (**d**) homocysteine (no methyl group) for *Pseudoalteromonas* sp. ASV16. Asterisks denote chemotactic index significantly larger than the filtered seawater controls (ANOVA (one-sided), $p$ < 0.05, all $p$-values are reported in Supplementary Data 23−25). Bars in a darker shade indicate a significantly stronger chemotactic response to laminarin compared to the controls without methionine, choline or homocysteine, respectively (ANOVA, $p$ < 0.05, all $p$-values are reported in Supplementary Data 23−25). Each treatment was replicated across three different ISCAs ($n$ = 3 biologically independent experiments, denoted by individual dots). Bar plots represent the mean ± SD. **e** Schematic of DMSP$_d$-enhanced chemotaxis to laminarin. In a given timeframe ($t_1$ to $t_2$), cells will cover larger distances along a laminarin gradient when DMSP$_d$ is present (left), than when DMSP$_d$ is absent (right). Our data suggest that this behaviour is induced because DMSP$_d$ acts as a precursor for methylation of the chemotaxis machinery, favoring the cell's adaptation to increasing polymer concentrations. Source data are provided as a Source Data file.

molarity or carbon content, as that would have resulted in far fewer molecules of laminarin. We therefore chose to employ the concentrations that elicited the strongest chemotactic response in laboratory conditions and in the field (Fig. S1). Yet only the highest concentration of laminarin induced a significant chemotactic response in laboratory conditions (Fig. S1a). We therefore chose to use 10 mg mL$^{-1}$ for laminarin. Laminarin at 10 mg mL$^{-1}$ did not alter the pH of the solution, as measured with an Orion Star 221 pH metre (ThermoFischer). In order to provide a similar number of molecules per substance, we used 1 mg mL$^{-1}$ for laminaribiose, 1 mg mL$^{-1}$ for glucose and 1 mg mL$^{-1}$ for mannitol. Assuming a polymer size range of 15–33 glucose molecules[19,20] (180 kDa each), a laminarin concentration of 10 mg mL$^{-1}$ (resulting in a molarity range of 1.8–4 mM) still contains on average 2 times less molecules than a solution of 1 mg mL$^{-1}$ glucose or

mannitol (5.5 mM). These working concentrations are representative of environmental hotspots of these chemicals[32]. Indeed, numerical modelling has revealed that the concentration of chemicals diffusing from the ISCA wells 1 mm away from the port mouth is nearly two orders of magnitude lower than the concentration in the well[32]. Thus, the signals created from the ISCA that are perceived by chemotactic microorganisms are of the same order of magnitude as the concentrations that arise from the lysis of a phytoplankton cell[61].

We confirmed the efficiency of the filtration procedure by flow cytometry and DNA sequencing. The resuspended chemoattractant solutions were filtered through a 0.2 µm Sterivex filter (Millipore) to remove undissolved particles. Samples of ultrafiltered seawater and resuspended chemoattractants were collected as controls for flow cytometry and DNA sequencing.

## Sample processing

**Flow cytometry.** Samples for flow cytometry were "flash thawed" at 40 °C, stained with SYBR Green I (1:10,000 final concentration; ThermoFisher), incubated for 15 min in the dark, and finally analysed on a CytoFLEX S flow cytometer (Beckman Coulter). Forward scatter (FSC), side scatter (SSC) and green (SYBR Green) fluorescence were recorded for each sample. Bacterial populations were characterised according to SSC and FITC (SYBR Green) and cell abundance for each sample was determined by analysing a standardised volume of sample (30 μL). The chemotaxis strength (or chemotactic index, $I_C$) was calculated by dividing the number of cells found in the chemoattractant wells by the number of cells found in the filtered seawater wells.

**DNA extraction, library preparation, sequencing and bioinformatics.** DNA extraction from ISCA samples was performed using a microvolume DNA extraction approach (physical lysis method)[62]. Libraries for 16S rRNA gene sequencing were generated for all samples using the Nextera XT DNA Sample Preparation Kit (Illumina) following a previously described protocol for low-input DNA libraries[63]. Primers targeting the v4-v5 regions of the 16S rRNA gene were used, namely 515 F (GTGYCAGCMGCCGCGGTAA) and 926 R (CCGYCAATTYMTT-TRAGTTT). Samples were then sequenced on an Illumina MiSeq platform. The sequencing run included a total of 69 samples: 29 ISCA samples from in situ deployments, 14 bulk seawater samples from the mesocosms and 24 controls (4 mock communities, 2 DNA extraction controls, 12 library prep controls, and 6 undeployed ISCA controls). These additional controls were used to identify and remove potential contaminants. Libraries were pooled on an indexed sequencing run and yielded ~400 bp per sample.

To characterise the composition of bacterial communities, reads were first trimmed with Seqtk[64] and pair-end were merged with FLASH[65]. Adaptors were removed using CutAdapt[66]. Denoising, dereplicating and taxonomy assignment were performed using DADA2[67]. Analysis of 16S rRNA gene-based taxonomic profiles was performed in the software environment R[68] with the phyloseq[69] package, and on the MicrobiomeAnalyst[70,71] platform. Data were rarefied to 1021 reads per samples.

To determine if statistical differences existed at the taxonomic level between treatments, we calculated Bray-Curtis similarity matrices on the relative abundance of rarefied reads. Permutational multivariate analyses of variance (PERMANOVA) were carried out using adonis function. Assessment of enriched taxa in laminarin treatments compared to the surrounding seawater was done using ANCOM-BC[72]. The 61 environmental variables were correlated with chemotactic indices induced by laminarin using Spearman's correlation (two-sided) with adjusted $p$-values. Data handling and production of graphs was performed using the following R packages: ggplot2, vegan, ggpubr, dplyr, tidyr, tibble.

**Purification and identification of field isolates.** To isolate chemotactic strains from each ISCA deployment, the 120 μL aliquots from the pooled wells of each chemoattractant from each ISCA were divided into six 20 μL aliquots and plated on 0.22-μm-filtered environmental seawater agar plates, amended with 1 mg mL$^{-1}$ of the corresponding chemoattractant. The two set of triplicate plates were incubated at room temperature and at fjord water temperature (10 °C) until colonies appeared on the surface. Individual colonies were then grown in 10% Difco 2216 Marine Broth (BD Diagnostics) in autoclaved seawater overnight at room temperature and at 10 °C. Each isolate went through two additional rounds of purification via successive growth on plate and liquid cultures. In total 186 strains were isolated during the filed campaign. Isolates were identified by 16S rRNA gene Sanger sequencing.

**Determination of DMSP$_d$ concentration.** For dissolved DMSP (DMSP$_d$) analysis, 10–15 mL of seawater were gravity filtered through a 25 mm GF/F filter, and the initial 3 mL of the filtrate were collected in a 10 mL glass vial. After addition of one NaOH pellet (45 mg, -0.1 mol l$^{-1}$ final concentration, pH > 12), the vial was crimped and stored overnight for alkaline DMSP hydrolysis. Evolved DMS was analysed by sparging the sample for 3–5 min with 40 mL min$^{-1}$ of high-purity helium (He), trapping the sparged volatiles at the temperature of liquid nitrogen, and re-volatilising them by dipping the loop in water at room temperature. Sulfur compounds were separated in a Shimadzu GC14A gas chromatograph using a packed CarbopackH 60/80 mesh column (Sigma-Aldrich) maintained at 170 °C, and detected by a flame photometric detector. Calibration was performed with a DMS standard solution prepared by alkaline hydrolysis of a DMSP solution in MilliQ water. The DMSP$_d$ concentration was calculated by subtraction of the corresponding, previously determined DMS concentration in the nonhydrolized sample.

## Laboratory experiments with environmental isolates

**Laminarin fractionation.** Laminarin (Sigma-Aldrich) was separated into size fractions using 3 kDa ultracentrifugal filter units (Amicon, Millipore) and 1 kDa dialysis membranes (Pur-A-Lyzer, Sigma-Aldrich) to give three fractions of polymers and oligomers: (i) >3 kDa, (ii) <3 kDa and (iii) 1–3 kDa.

**Bacterial cell culture.** *Alteromonas* sp. (ASV109 and ASV76) and *Pseudoalteromonas* sp. (ASV39 and ASV16), isolated from laminarin-containing ISCA wells during the field deployments, were used in the laboratory-based experiments. The cells were plated on 100% Difco 2216 Marine Broth (BD Diagnostics) and grown overnight at 30 °C. The following day, one single colony was inoculated in 100 mL artificial seawater (Instant Ocean, Spectrum Brands) amended with 10 mL Difco 2216 Marine Broth medium (BD Diagnostics). The flasks were grown at 30 °C for 16 h in a shaking incubator. Before experiments, bacterial motility was confirmed by microscopy and the cell concentration was determined by flow cytometry.

**Chemotaxis experiments with laminarin and alginate, and constituent oligomers and monomers.** We performed experiments to quantify attraction of the four environmental bacteria isolates to the following chemoattractants: commercially available laminarin extract (10 mg mL$^{-1}$, Sigma-Aldrich), three individual laminarin fractions derived from the solution of raw laminarin (10 mg mL$^{-1}$, Sigma-Aldrich), laminarin hexose (1 mg mL$^{-1}$, Megazyme), laminaribiose (1 mg mL$^{-1}$, Megazyme), glucose (1 mg mL$^{-1}$, Sigma-Aldrich), alginate solution (1 mg mL$^{-1}$, Sigma-Aldrich), alginate oligomers (1 mg mL$^{-1}$, Sigma-Aldrich) and mannuronate (1 mg mL$^{-1}$, Sigma-Aldrich).

Further tests were carried out comparing 1 mg mL$^{-1}$ laminarin and 1 mg mL$^{-1}$ glucose to confirm that the observed chemotactic responses were not due to concentration effects (Fig. S11 and Supplementary Data 27). Additionally, ASV39 was employed to compare the chemotactic response induced by 10 mg mL$^{-1}$ and 1 mg mL$^{-1}$ of glucose. In both cases, no significant response was measured when compared to artificial seawater (Fig. S12, $p > 0.05$ for both concentration, Supplementary Data 28).

**Chemotaxis experiments in the presence of DMSP, methionine, choline or homocysteine in the surrounding seawater.** Stock solutions (10 mM) of DMSP, methionine, choline and homocysteine were prepared in autoclaved artificial seawater (Instant Ocean, Spectrum Brands) and diluted in individual flasks containing 250 mL of the same artificial seawater to final concentrations of (i) 100 nM, (ii) 1 μM and (iii) 10 μM to simulate concentrations found naturally within the phycosphere surrounding phytoplankton cells in the blooms[6]. Cell counts of each culture of the four environmental isolates were determined by flow cytometry and individually diluted in each flask to obtain a cell suspension of 10$^6$ cells mL$^{-1}$. The cells were incubated in the flasks for

15 min at room temperature. During that time, ISCA triplicates for each concentration of the amended chemical were loaded with the test chemicals: laminarin (10 mg mL$^{-1}$, Sigma-Aldrich), the <3 kDa fraction derived from an original laminarin solution of 10 mg mL$^{-1}$, laminaribiose (1 mg mL$^{-1}$, Megazyme), glucose (1 mg mL$^{-1}$, Sigma-Aldrich), alginate (0.1%, Sigma-Aldrich), alginate oligomers (0.1%, Sigma-Aldrich), and mannuronate (1 mg mL$^{-1}$, Sigma-Aldrich). Additional tests in the presence of DMSP were run using aspartate (1 mM, Sigma-Aldrich), serine (1 mM, Sigma-Aldrich), spermidine (1 mM, Sigma-Aldrich), trimethylamine (TMA, 1 mM, Sigma-Aldrich) and 10% Marine Broth 2216 (Sigma-Aldrich). A row of wells containing artificial seawater acted as a negative control on all ISCAs. The devices were then placed in sterile artificial seawater containing the specified concentration of DMSP. A fourth set of triplicate ISCAs were prepared with the same chemoattractants but exposed to a bacterial culture lacking any DMSP. The ISCAs were incubated for 1 h in the bacterial cultures and then the contents of the wells were retrieved and the cells counted by flow cytometry after SYBR Green I staining (ThermoFisher).

For the chemotaxis experiment with *E. coli* K-12 strain RP437, the cells were grown overnight in Terrific Broth (Sigma-Aldrich) and then subcultured in the same medium for 4 h until reaching an OD = 0.45. The culture was then washed twice and resuspended in methionine-free Motility Buffer[10]. The chemotaxis assay was performed as described above, using 1 mM aspartate as chemoattractant and methionine or DMSP as amending chemicals. Cells were diluted 1/100 in motility buffer containing 0.1, 1 and 10 µM methionine, and in parallel, the same concentrations of DMSP. All ISCAs also contained a row of motility buffer as negative control.

**Chemotaxis assay to DMSP.** In order to test whether DMSP itself induced chemoattraction in our four marine strains, ISCA experiments were conducted with the same concentration range used in the DMSP-addition laboratory experiments (0.1–10 µM; Fig. S13). A stock solution (10 mM) of DMSP was prepared in autoclaved artificial seawater (Instant Ocean, Spectrum Brands) and diluted in individual 15 mL tubes (Falcon) containing 10 mL of the same artificial seawater to final concentrations of (i) 0.1 µM, (ii) 1 µM and (iii) 10 µM. Cell counts of each overnight culture of the four environmental isolates were determined by flow cytometry and cell cultures were diluted in each flask to obtain a cell suspension of 10$^6$ cells mL$^{-1}$. For each isolate, three ISCAs were each loaded with the three concentrations of DMSP, with one row of wells containing artificial seawater as a negative control. The ISCAs were incubated for 1 h in the diluted bacterial cultures. Thereafter, the contents of the wells were retrieved and the cells counted by flow cytometry after SYBR Green I staining (ThermoFisher). The chemotactic index was determined from the cell counts as described in "*Sample processing - Flow cytometry*".

**Growth assays.** Overnight cultures of the four environmental isolates grown in 10% Difco 2216 Marine Broth medium (BD Diagnostics) were washed twice (160 rcf, 15 min) in artificial seawater (Instant Ocean, Spectrum Brands) and diluted 1/100 in 10 mL tubes containing artificial seawater amended with (i) 1 mg mL$^{-1}$ laminarin, (ii) 1 mg mL$^{-1}$ laminaribiose, (iii) 1 mg mL$^{-1}$ glucose, or (iv) no amendment as negative control. All treatments were conducted in triplicate. Cultures were maintained at 27 °C in a shaking incubator and sampled at 0 h, 24 h and 48 h for cell quantification via flow cytometry. To characterise growth, cell counts at 24 h and 48 h were normalised by the initial cell count and are represented as a fold increase.

To confirm that the cell number in the ISCA wells upon amendment of DMSP or laminarin was not due to the growth of the bacterial isolates, we ran growth assays replicating how cells are prepared in ISCA experiments. Overnight cultures of the four environmental isolates were grown in 10% Difco 2216 Marine Broth medium (BD Diagnostics) and were subsequently diluted 1/100 in 10 mL tubes

containing artificial seawater amended with (i) 10 mg mL$^{-1}$ laminarin, (ii) 10 µM DMSP or (iii) no amendment as negative control. All treatments were conducted in triplicate. Bacterial growth was measured at 27 °C in a shaking plate reader for 48 h.

**Cell tracking.** Overnight cultures of *Pseudoalteromonas* sp. ASV16 and ASV39 grown in 10% Difco 2216 Marine Broth medium (BD Diagnostics) were diluted 1/200–1/500 in artificial seawater (Instant Ocean, Spectrum Brands) containing (i) 10 µM DMSP, (ii) 1 µM laminarin, (iii) 10 µM DMSP and 1 µM laminarin or (iv) no amendment as negative control. At regular intervals after dilution (15–75 min), 45 µL of cell suspensions were placed in the centre of a chamber (created by fixing a coverslip on a standard microscopy slide separated by ~1 mm of silicone rubber) and observed using phase contrast microscopy (Nikon) with a 20 × 0.45 NA objective (Nikon). Videos with acquisition speed 25–30 frames per second were recorded using a CMOS camera (Hamamatsu) for 30 s, with a resolution of 2044 × 2048 pixels (0.326 µm/pixel). Cell tracking was performed using TrackPy[73,74] (v 0.5.0) after removing the background from each image by substracting the median image computed over the entire video. In the analyses, we allowed a maximum displacement per frame of 31 pixels (corresponding to ~200 µm s$^{-1}$) and minimum separation between particles of 51 pixels. Trajectories were corrected for drift and cell positions were averaged over a time window of 9 frames in our calculation of the speed. Cells with an average speed of less than 12 µm/s were classified as non-motile. For the calculation of the reorientation frequency, we used a similar approach as published previously[75]. The cellular positions processed with a second-order Savitzky-Golay filter[76] with a time window of 5 frames and the change in angle was computed from the filtered positional data. For each trajectory, time points where both (A) the minimal absolute change in angle exceeded 25° and (B) the filtered velocity was lower than 70% of the average velocity of the trajectory, were marked as reorientation events (Fig. S7). Minimal time between two reorientation events was limited to 2 frames (60–80 ms). The run time was defined as the time between detected reorientation events. The first run (from the start of the trajectory to the first event) and the last run (from the last detected event to trajectory length) were used as lower bounds estimates of the run time. The reorientation frequency per cell was calculated as the inverse of the mean average run time per cell.

**In vitro enzyme assays.** In order to identify the mechanism by which the addition of DMSP affects chemotaxis, we performed an in vitro enzyme assay to determine which substrates induce the production of methionine by *Pseudoalteromonas* sp. ASV16 cell lysate. *Pseudoalteromonas* sp. ASV16 cells were grown overnight in rich medium (10% Marine Broth 2216; Sigma-Aldrich) and resuspended in artificial seawater (Instant Ocean, Spectrum Brands) at 10$^6$ cells mL$^{-1}$ in a 50 mL tube (Falcon). Three replicate experiments were performed. DMSP (10 µM, Sigma-Aldrich) and laminarin (10 mg mL$^{-1}$, Sigma-Aldrich) were added to the cell cultures, which were then incubated for 1 h at room temperature, to mimic the conditions of an ISCA experiment. After this incubation period, the cell cultures were spun down (700 rcf, 20 min) and resuspended in artificial seawater. This washing step was repeated a second time and the cell pellet was then resuspended in 250 µL of MilliQ water containing Roche Complete protease inhibitor (Sigma-Aldrich) before it was snap-frozen in liquid nitrogen and stored at −80 °C until the assay was performed. Before use, the resuspended cells were thawed at room temperature for 30 min to promote cell lysis, then briefly vortexed and transferred to ice for another 15 min. The cells were then centrifuged at 16,000 rcf for 5 min and the supernatant was collected and used as the protein extract for the in vitro enzyme assay. DMSP methyltransferase activity was tested using an in vitro enzyme assay, using 200 µM of DMSP and homocysteine in four different combinations: DMSP only, homocysteine only, both, and none of the

two. Each substate combination was diluted in 20 mM of ammonium bicarbonate buffer (pH 7.8) in triplicate treatments. The assay was initiated by adding 10 μL of the protein extract to 90 μL of the enzyme assay mix. The enzyme assay was performed in the autosampler of an Agilent 1290 Infinity LC stack kept at 18 °C and formation of methionine was monitored by sampling the enzyme reaction continuously over time. Measurements were performed using Liquid Chromatography coupled with an Agilent 6546 Quadrupole Time of Flight Mass Spectrometer in positive mode, 10 GHz, high-resolution mode. An Agilent EC-CN Poroshell column (50 mm × 2.1 mm, 2.7 μM) was used isocratically to reduce interference of salts on metabolite ionisation[77]. The buffer consisted of 10% acetonitrile (CHROMASOLV) in 90% water with 0.1% formic acid (Sigma-Aldrich), with a flow rate of 1 mL min$^{-1}$ at 20 °C. Every 2 min, a 3 μL sample was injected into the instrument. Raw data was treated with a spectral processing and alignment pipeline[78] using Matlab (The Mathworks, Natick).

**Orthology of DMSP catabolism genes.** To determine whether genes involved in DMSP catabolism are present in our bacterial strains, the 16S rRNA gene sequences of the four isolates were used to identify genomes of the closest sequenced organisms using BLAST$_n$ on the KEGG database[79] (minimum similarity: 98.02%). Each DMSP catabolism gene (*dmdA*, *dmdB*, *dmdC*, *dmdD*, *dddD*, *dddP*, *dddY*, *dddQ*, *dddW* and *dddL*) was then queried in the KEGG database[79] and the sequences harboured by Gammaproteobacterial genomes were selected. Each DMSP-degrading gene was used in a BLAST$_p$ analysis (KEGG[79]) to search for orthologous sequences in the closest relatives of our marine isolates (Supplementary Data 26). Finally, a reciprocal best hits BLAST of the identified orthologous sequences was carried out in the genomes of the Gammaproteobacteria from which these DMSP-degrading genes originated (Supplementary Data 26).

**Reporting summary**
Further information on research design is available in the Nature Portfolio Reporting Summary linked to this article.

## Acknowledgements
We thank S. Kobel and A.M. Minder Pfyl (Genetic Diversity Center, ETH Zurich, Switzerland) for their help and advice in the 16S rRNA gene library preparation and sequencing. We thank R. Pioli (ETH Zurich, Switzerland) for the illustration used in Fig. 4e. "We thank M.A. Moran for fruitful discussions on DMSP demethylation". We further thank all participants of the AQUACOSM VIMS-EHUX project who measured the environmental variables used for the correlation analyses (from the A. Vardi, R. Simò, O. Cordero, B. Bailleul, J.H. Hehemann, J.K. Egge, A. Larsen labs). We gratefully acknowledge financial support from Australian Research Grant DP200100919 to J-B.R., R.S. and J.R.S.; and a Swiss National Science Foundation grant 205321_207488, a Gordon and Betty Moore Foundation Symbiosis in Aquatic Systems Initiative Investigator Award (GBMF9197; https://doi.org/10.37807/GBMF9197), and the Simons Foundation through the Principles of Microbial Ecosystems (PriME) collaboration (grant 542395) to R.S. J-B.R. was supported by an Australian Research Council Future Fellowship (FT210100100). R.Si. and M.M.N. were supported by the European Research Council (ERC) under the EU Horizon2020 program (ERC-2018-ADG #834162 to R.Si.), and the Spanish MICINN through the BIOGAPS grant (CTM2016-81008-R) to R.Si. and the "Severo Ochoa Centre of Excellence" accreditation (CEX2019-000298-S) to the ICM. The mesocosm experiment VIMS-Ehux was supported by EU Horizon2020-INFRAIA project AQUACOSM (grant no. 731065).

## Author contributions
E.E.C., J.-B.R., J.M.K., U.A., Z.L., J.R.S. and R.S. designed the experiments. E.E.C., J.-B.R. and B.S.L. performed in situ experiments and strain isolation. E.E.C., J.M.K., U.A., V.A. and F.D.S. performed the laboratory experiments. E.E.C., J.-B.R. and Z.L. generated and analysed the amplicon data. S.P designed enzyme assay experiments and performed the mass spectrometry measurements and data analysis. U.S. provided oversight on chemical measurements by Mass Spectrometry. F.V., M.M.N., A.V., A.S., R.Si. and J.-H.H. measured environmental variables (phytoplankton counts, DMSP and glycans concentrations and all other environmental bio-geochemical and physical variables). E.E.C., J.-B.R., J.M.K., J.R.S. and R.S. wrote the manuscript. All authors edited the manuscript before submission.

## Competing interests
The authors declare no competing interests.

## Data availability
The amplicon sequencing data (16S rRNA gene) of the four isolates have been deposited on NCBI under accession numbers: OR501448-51 [https://www.ncbi.nlm.nih.gov/nuccore/OR501448.1/]; [https://www.ncbi.nlm.nih.gov/nuccore/OR501449.1/]; [https://www.ncbi.nlm.nih.gov/nuccore/OR501450.1/]; [https://www.ncbi.nlm.nih.gov/nuccore/OR501451.1/]. Raw amplicon fastq files were deposited in the Sequence Read Archive under accession number PRJNA1015554]. Raw spectral files for in vitro enzyme assays have been deposited into the MassIVE database, with accession code MSV000092825 [https://doi.org/10.25345/C5VD6PF94]. Source data are provided with this paper.

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
