## [Peer Review file · Nature Communications]

Editorial Note: Parts of this Peer Review File have been redacted as indicated to remove third-party material where no permission to publish was obtained.

Reviewers' Comments:

Reviewer #1:

Remarks to the Author:

Laminarin is an abundant algal polysaccharide composed of 15 to 33 glucose monomers. It is a carbon storage compound for a range of micro- and macroalgae. Laminarin is released upon cell lysis and metabolized by marine microorganisms. Using in situ chemotaxis assays to 10 mg/ml laminarin, the authors show that many marine bacteria are attracted by laminarin. Laminarin is a heterogeneous polysaccharide and the authors show that the higher molecular forms are particularly strong chemoattractants. It was also shown that chemotactically attracted bacteria are able to use laminarin for growth. Based on the observation that the magnitude of laminarin chemotaxis varied during plankton bloom, the authors conducted studies to identify the signals that may be responsible for this variation. These experiments resulted in the identification of dimethylsulfoniopropionate (DMSP), a compound mainly produced by phytoplankton. When chemotaxis experiments with artificial seawater were conducted using laminarin, spiking with DMSP enhanced chemotaxis. In the last section of their work the authors conducted experiments to identify the molecular mechanism of the DMSP action. They conclude that DMSP enhances the cellular capacity to methylate chemoreceptors similar to that seen for methionine (the precursor for the synthesis of S-adenosylmethionine, the substrate of the CheR methyltransferase). This is a solid and rigorously conducted study that further underlines the very important role chemotaxis has in marine organisms. The manuscript is written in a precise and clear manner. At a number of places, derived conclusions and hypothesis could be further supported by experimental data. The last section on the molecular mechanism of DMSP requires further experiments if the authors want to maintain their hypothesis.

Major points:

1. The authors used a laminarin concentration of 10 mg/ml and conducted spiking experiments with 0.1 to 10 micromolar DMSP. Certainly, in situ gradients will be of short-scale and locally around algae. However, this manuscript would largely benefit from data illustrating that responses to these concentrations are of physiological relevance. Ideal would be chemotactic measurements towards algae that release laminarin and mutant derivatives that are unable to produce this compound.
2. There are already a number of reports on chemotaxis to large polysaccharides. This contrasts somewhat with our current understanding on the molecular detail of signal sensing by chemoreceptor, that consists of sensor domains that possess (chemoreceptors as well as solute binding proteins) well-defined binding pockets for low molecular weight compounds. Certainly, it is not the objective of this article to identify the molecular mechanism of laminarin chemotaxis. However, since the authors hypothesize that laminarin enters the periplasm, the authors are invited to present data that support this hypothesis. It is relatively straightforward to produce an extract of periplasmic compounds. A demonstration that the periplasmic concentration of laminarin increases upon exposure to laminarin, would provide further support to the hypothesis brought forward by the authors.
3. Have the authors assessed whether laminarin is a pH-active compound, i.e. the addition of laminarin alters the pH of seawater?
4. The least mature part of the manuscript is the section on the molecular mechanism of DMSP. The authors issue the rather vague statement that "that DMSP enhances chemotaxis towards laminarin by supplying methyl groups to MCPs". Do the authors suggest that DMSP performs CheR independent chemoreceptor methylation or that CheR uses DMSP as methylation substrate instead of SAM? Chemoreceptor methylation is carried out by the CheR methyltransferase that uses SAM as substrate producing SAH. Data available show that the product SAH binds with much higher affinity than SAM to CheR exerting product feedback inhibition of methylation. In addition, there are several enzymes that degrade cellular SAH and their activity is also regulated. CheR methylation activity thus is dependent on the concentration and above all the SAM/SAH ratio. If the authors want to maintain their hypothesis on the molecular mechanism of DMSP, they may want to determine by quantitative mass spectrometry approaches SAM and SAH concentrations and ratios of both compounds in cells in the presence and absence of DMSP.
5. Is DMSP a chemoattractant? Within the chemotactic array there are cooperative interactions between neighboring receptors that recognize different ligands.

Reviewer #2:

Remarks to the Author:

This study reports chemotaxis to polysaccharide laminarin and alginate by marine bacteria and enhancement of their chemotaxis by DMSP by serving as methyl donor to MCPs. Finding of chemotaxis to high molecular weight compounds itself is very interesting because chemotaxis towards compounds with low molecular weights have been mainly investigated so far. Actually, findings in this paper will contribute to understanding carbon cycle in marine environments.

Specific comments:

1. Are polysaccharides themselves chemotactic ligands? It is possible that metabolites derived from laminarin are chemotactic ligands. It is interesting to quantify responses toward culture supernatant on medium containing laminarin.
2. When DMSP stimulates chemotaxis to polysaccharides by serving as methyl donor to MCPs in marine bacteria, it also stimulates chemotaxis toward chemicals with low molecular weights (like chemotaxis to aspartate in *E. coli*). Does DMSP stimulate chemotaxis of marine bacteria to chemicals with low molecular weights (for example, amino acids or casamino acids)? I consider the possibility that DMSP enhances uptake of polysaccharides, leading to enhancement of chemotaxis to polysaccharides (if it is correct, DMSP enhances growth of marine bacteria on laminarin).

Junichi Kato (Hiroshima University)

Reviewer #3:

Remarks to the Author:

Clerc et al. demonstrated that laminarin can elicit strong chemoattraction, which can be further enhanced by the important organic sulfur compound DMSP, by their in situ chemotaxis assay (ISCA). This work for the first time indicated a potential role of DMSP in promoting bacterial chemoattraction, which could be an important finding. The authors also provide an explanation of the possible mechanism behind this role of DMSP. The manuscript is generally well-written. However, in my opinion, some questions remained to be answered in this manuscript regarding the DMSP related results, which might make their conclusions more reliable.

Major:

1. DMSP could promote chemoattraction is the most important conclusion of this paper. However, it seems that the authors do not fully describe the importance of DMSP in their introduction, especially about its role as a chemical cue in previous publications. This information has only been mentioned briefly in the first paragraph and later in the results and discussion, which should be better to expand this somewhere appropriate. I think this information is necessary for the readers to understand the importance of this paper.
2. Line 173-183: Can you rule out the possibility that the observed higher I_c of laminarin is affected by the effect related to the stimulated growth?
Moreover, when you discussed about the chemoattraction promotion effect of DMSP (in line 250-252), have you ruled out the possibility that DMSP may promote the bacterial growth since it is an important carbon source?
3. Many marine bacteria are involved in DMSP degradation and producing DMS or MeSH. Have the authors confirmed that their *Pseudoalteromonas* and *Alteromonas* strains were able to catabolize DMSP or not? Otherwise there might also be the influence of resulting DMS or MeSH? As far as we know at least some of isolates in these two genera have such capacity.
4. DMSP itself has been reported to stimulate chemotactic responses. Have you tested that the chemoattraction effect of DMSP in your ISCA?
5. The authors assumed that DMSP acts as a source of methyl groups. However, it seems that there are no publications about methyltransferases specific on DMSP. It is not like methionine, whose methyltransferases were widely existed in diverse bacteria. Although the authors provided indirect evidence based on the experiment with other compounds such as choline, it would be better if the authors can provide more direct evidence. For example, how could methyl transfer happen with DMSP or what is the resulting product?

Response to Reviewer 1

For ease of editorial review, we have included the Reviewers' comments below in black font. Our responses are in blue and *italics*. Text that has been inserted or changed within the modified manuscript is underlined.

Laminarin is an abundant algal polysaccharide composed of 15 to 33 glucose monomers. It is a carbon storage compound for a range of micro- and macroalgae. Laminarin is released upon cell lysis and metabolized by marine microorganisms. Using *in situ* chemotaxis assays to 10 mg mL⁻¹ laminarin, the authors show that many marine bacteria are attracted by laminarin. Laminarin is a heterogeneous polysaccharide and the authors show that the higher molecular forms are particularly strong chemoattractants. It was also shown that chemotactically attracted bacteria are able to use laminarin for growth. Based on the observation that the magnitude of laminarin chemotaxis varied during plankton bloom, the authors conducted studies to identify the signals that may be responsible for this variation. These experiments resulted in the identification of dimethylsulfoniopropionate (DMSP), a compound mainly produced by phytoplankton. When chemotaxis experiments with artificial seawater were conducted using laminarin, spiking with DMSP enhanced chemotaxis. In the last section of their work the authors conducted experiments to identify the molecular mechanism of the DMSP action. They conclude that DMSP enhances the cellular capacity to methylate chemoreceptors similar to that seen for methionine (the precursor for the synthesis of S-adenosylmethionine, the substrate of the CheR methyltransferase).

This is a solid and rigorously conducted study that further underlines the very important role chemotaxis has in marine organisms. The manuscript is written in a precise and clear manner. At a number of places, derived conclusions and hypothesis could be further supported by experimental data. The last section on the molecular mechanism of DMSP requires further experiments if the authors want to maintain their hypothesis.

We thank the Reviewer for their accurate summary and their appraisal of our work. We have conducted multiple additional experiments to address their remarks, which we describe in detail below.

Major points

1.1. The authors used a laminarin concentration of 10 mg mL⁻¹ and conducted spiking experiments with 0.1 to 10 μM DMSP. Certainly, *in situ* gradients will be of short-scale and locally around algae. However, this manuscript would largely benefit from data

illustrating that responses to these concentrations are of physiological relevance. Ideal would be chemotactic measurements towards algae that release laminarin and mutant derivatives that are unable to produce this compound.

*We thank the Reviewer for this suggestion. The genomic pathway involved in the biosynthesis of laminarin in algae is, to the best of our knowledge, incompletely described (Michel et al., 2010), unfortunately preventing the use of targeted mutagenesis. However, we can assert that the concentrations of both laminarin and DMSP used in our experiments are of environmental relevance. Indeed, the concentrations of DMSP in the bulk seawater during the *Emiliana huxleyi* bloom in our field experiments reached 0.1 μM , which is equal to the lowest concentration used in our DMSP-amendment experiments. It is also important to highlight that concentrations of DMSP in the direct vicinity of producing cells, or when released upon cell lysis, can reach concentrations 6-7 orders of magnitude higher (hundreds of mM) than those in the average background seawater (Caruana and Malin, 2014), far exceeding the highest concentrations we employed in our experiments.*

To clarify this point in the manuscript, we have added the following to the main text:

“We quantified their chemotactic response towards laminarin (10 mg mL⁻¹) when the background artificial seawater (outside of the ISCA wells) was spiked with DMSP_d at different concentrations (0.1, 1 and 10 μM , Supplementary Note 2), representative of phytoplankton lysis events (Caruana and Malin, 2014), yet not sufficient to induce bacterial growth within 1h (Fig. S5, Table S18).”

Additionally, while the concentration of laminarin tested in the ISCA might seem high (10 mg mL⁻¹), a sharp concentration gradient occurs between the inside of an ISCA well and the outside seawater where chemotactic microbes are recruited. Indeed, mathematical modeling has demonstrated that concentrations of chemicals diffusing from an ISCA well (i.e., the concentrations sensed by chemotactic microbes in the seawater outside of the device) are 1-2 orders of magnitude lower than the concentration within the well (Lambert et al., 2017; see figure below). According to our calculations¹, the concentrations of laminarin in the bulk seawater during our field experiment reached values of the order of 0.25 mg mL⁻¹, i.e., within the same order of magnitude as the concentrations produced by diffusion from the ISCA wells.

¹ The maximum particulate organic carbon concentrations during the *E. huxleyi* bloom was 138.2 $\mu\text{mol L}^{-1}$. Since laminarin can represent up to 42% of POC during phytoplankton blooms (Becker et al., 2020), this indicates that laminarin may have accounted for 58 $\mu\text{mol L}^{-1}$ of carbon, which corresponds to 0.25 mg mL⁻¹ of laminarin for average-size polymers (4320 Da).

[redacted]

Figure adapted from Lambert et al (2017). a) Predicted concentration field of a chemoattractant after 1 h. Cross-section for an individual ISCA well taken from a COMSOL model of diffusion. Red dashed lines correspond to port entrance and exit heights and the yellow dashed line along the gradient indicates the location of the transect used to generate the data in panel (b). b) Rescaled concentration profile along the height of the well, the inlet and the region directly above the inlet. Red dashed lines as in panel a. Note that this figure was originally produced for alpha-methylaspartate, but the conclusion that there is a sharp decrease in concentration from within to immediately outside the ISCA well applies to all compounds.

We have added further explanation in the Methods section to cover this point, as follows:

“These working concentrations are representative of environmental hotspots of these chemicals (Lambert et al., 2017). Indeed, numerical modelling has revealed that the concentration of chemicals diffusing from the ISCA wells 1 mm away from the port mouth is nearly two orders of magnitude lower than the concentration in the well (Lambert et al., 2017). Thus, the signals created from the ISCA that are perceived by chemotactic microorganisms are of the same order of magnitude as the concentrations that arise from the lysis of a phytoplankton cell (Stocker and Seymour, 2012).”

1.2. There are already a number of reports on chemotaxis to large polysaccharides. This contrasts somewhat with our current understanding on the molecular detail of signal sensing by chemoreceptor, that consists of sensor domains that possess (chemoreceptors as well as solute binding proteins) well-defined binding pockets for low molecular weight compounds. Certainly, it is not the objective of this article to identify the molecular mechanism of laminarin chemotaxis. However, since the authors hypothesize that laminarin enters the periplasm, the authors are invited to present data that support this hypothesis. It is relatively straightforward to produce an extract of periplasmic compounds. A demonstration that the periplasmic concentration of

laminarin increases upon exposure to laminarin, would provide further support to the hypothesis brought forward by the authors.

We thank the Reviewer for this comment. The uptake of entire molecules of laminarin in the periplasm has already been reported several times in the literature (Unfried et al., 2018; Konishi et al., 2020; Reintjes et al., 2017; Mystkowska et al., 2018). We were therefore not intending to formulate a new hypothesis, but simply to link our results to previous findings. To clarify this point, we have modified the text and it now reads:

“In contrast, polysaccharides have been shown to first require active transport into the periplasm by transmembrane proteins in order to be sensed by cells (Unfried et al., 2018; Konishi et al., 2020; Reintjes et al., 2017; Mystkowska et al., 2018). The strong response of three of our isolates (ASV16, ASV39 and ASV76) to the largest polymer size could therefore indicate whole polymer uptake and sensing, possibly mediated by the SusD-binding protein (Mystkowska et al., 2018).”

1.3. Have the authors assessed whether laminarin is a pH-active compound, i.e. the addition of laminarin alters the pH of seawater?

Laminarin has been reported to be a neutral polysaccharide in the literature (Rinaudo, 2007). We confirmed this experimentally in our work, finding that the addition of 10 mg mL⁻¹ (the highest concentration we have used in chemotaxis experiments) of laminarin to filtered artificial seawater did not change the pH of the solution. We now state this within the Methods section:

“Laminarin at 10 mg mL⁻¹ did not alter the pH of the solution, as measured with an Orion Star 221 pH meter (ThermoFisher).”

1.4. The least mature part of the manuscript is the section on the molecular mechanism of DMSP. The authors issue the rather vague statement that “that DMSP enhances chemotaxis towards laminarin by supplying methyl groups to MCPs”. Do the authors suggest that DMSP performs CheR independent chemoreceptor methylation or that CheR uses DMSP as methylation substrate instead of SAM? Chemoreceptor methylation is carried out by the CheR methyltransferase that uses SAM as substrate producing SAH. Data available show that the product SAH binds with much higher affinity than SAM to CheR exerting product feedback inhibition of methylation. In addition, there are several enzymes that degrade cellular SAH and their activity is also regulated. CheR methylation activity thus is dependent on the concentration and above all the SAM/SAH ratio. If the authors want to maintain their hypothesis on the molecular mechanism of DMSP, they may want to determine by quantitative mass spectrometry

approaches SAM and SAH concentrations and ratios of both compounds in cells in the presence and absence of DMSP.

We thank the Reviewer for prompting us to conduct further experiments to elucidate the mechanism of methyl transfer, which we now successfully did. It is important to clarify that we did not mean to suggest that CheR binds to DMSP instead of SAM, but instead that DMSP acts as a methyl donor for synthesis of methionine or SAM. Our ISCA experiments show that DMSP and methionine, but not homocysteine, increase the strength of chemotaxis, but in our original manuscript we did indeed not provide direct evidence that DMSP can be converted into methionine (and ultimately SAM). We have now performed mass spectrometry (MS) measurements, as suggested by the Reviewer, and we are delighted to provide direct evidence that DMSP can provide a methyl group into the methionine-SAM pathway.

Specifically, we conducted in vitro enzyme assays (see Methods) employing Pseudoalteromonas sp. ASV16 cell lysate as the enzymatic source. In this assay, we monitored the production of methionine using MS when we exposed the cell lysate to different molecules. We observed an increase in methionine levels only when both homocysteine and DMSP were added to the extract in combination (Fig. S10d, see below). These results provide direct evidence that ASV16 possesses enzymes that perform methyl transfer from DMSP to homocysteine to form methionine. Methionine can then be converted to SAM (Cantoni, 1951; Armstrong, 1972; Lu, 2000), which binds to CheR and enhances chemotaxis (Adler, 1973).

We have modified the main text to include these results, as follows (note that Fig. S10 is reported also here below for convenience):

“Finally, using an in vitro enzyme assay employing Pseudoalteromonas sp. ASV16 cell lysate as an enzymatic source, we demonstrated that the cell lysate is capable of producing methionine only when supplied with DMSP and homocysteine in combination (Fig. S10d). These results provide direct evidence that ASV16 possesses enzymes able to transfer a methyl group from DMSP to homocysteine to form methionine. This reaction is likely mediated through a yet uncharacterized pathway, as all close relatives of the marine strains used here lack the dmdA gene, the only gene known to be capable of demethylating DMSP (Reisch et al. 2011; Table S26).”

In order to clarify that the conversion of methionine into SAM is a common metabolic process in bacteria, we have also added the following statement:

“These studies revealed that the sensitivity of chemoreceptors (methyl-accepting chemotaxis proteins, MCPs) in E. coli strongly depends on the availability of methyl groups provided by S-adenosyl-methionine (SAM), a methyl donor that is derived from

methionine (Armstrong, 1972), which occurs in high intracellular concentration in bacteria and is involved in many cellular processes (Cantoni, 1951; Armstrong, 1972; Lu, 2000)."

Further, we have added the enzyme assay in the Methods section, which now reads:

"In order to identify the mechanism by which the addition of DMSP affects chemotaxis, we performed an in vitro enzyme assay to determine which substrates induce the production of methionine by Pseudoalteromonas sp. ASV16 cell lysate. Pseudoalteromonas sp. ASV16 cells were grown overnight in rich medium (10% Marine Broth 2216; Sigma-Aldrich) and resuspended in artificial seawater (Instant Ocean, Spectrum Brands) at 10^6 cells mL^{-1} in a 50 mL tube (Falcon). Three replicate experiments were performed. DMSP (10 μM , Sigma-Aldrich) and laminarin (10 mg mL^{-1} , Sigma-Aldrich) were added to the cell cultures, which were then incubated for 1 h at room temperature, to mimic the conditions of an ISCA experiment. After this incubation period, the cell cultures were spun down (3000 rpm, 20 min) and resuspended in artificial seawater. This washing step was repeated a second time and the cell pellet was then resuspended in 250 μL of MilliQ water containing Roche Complete protease inhibitor (Sigma-Aldrich) before it was snap-frozen in liquid nitrogen and stored at $-80^\circ C$ until the assay was performed.

Before use, the resuspended cells were thawed at room temperature for 30 min to promote cell lysis, then briefly vortexed and transferred to ice for another 15 min. The cells were then centrifuged at 10,000 rpm for 5 min and the supernatant was collected and used as the protein extract for the in vitro enzyme assay. DMSP methyltransferase activity was tested using an in vitro enzyme assay, using 200 μM of DMSP and homocysteine in four different combinations: DMSP only, homocysteine only, both, and none of the two. Each substrate combination was diluted in 20 mM of ammonium bicarbonate buffer (pH 7.8) in triplicate treatments. The assay was initiated by adding 10 μL of the protein extract to 90 μL of the enzyme assay mix. The enzyme assay was performed in the autosampler of an Agilent 1290 Infinity LC stack kept at $18^\circ C$ and formation of methionine was monitored by sampling the enzyme reaction continuously over time.

Measurements were performed using Liquid Chromatography coupled with an Agilent 6546 Quadrupole Time of Flight Mass Spectrometer in positive mode, 10 GHz, high resolution mode. An Agilent EC-CN Poroshell column (50 mm \times 2.1 mm, 2.7 μM) was used isocratically to reduce interference of salts on metabolite ionization (Pontrelli and Sauer, 2021). The buffer consisted of 10% acetonitrile (CHROMASOLV) in 90% water with 0.1% formic acid (Sigma-Aldrich), with a flow rate of 1 $mL\ min^{-1}$ at $20^\circ C$. Every 2 min, a 3 μL sample was injected into the instrument. Raw data was treated with a

spectral processing and alignment pipeline using Matlab (The Mathworks, Natick) as described previously (Fuhrer et al., 2011)."

All metabolomics raw spectral files have been deposited into the MassIVE database with the accession code MSV000092825 and password "reviewer123". The dataset will be made publicly accessible upon acceptance.

Finally, the Reviewer suggested quantifying SAM/SAH ratios. We thank the Reviewer for this constructive suggestion and we agree that this assay in theory would also have provided the required biochemical evidence. However, we chose to focus on the in vitro enzymatic assay as we believed this was the critical missing biochemical step that was not supported by literature, and we anticipated conceptual and experimental challenges in obtaining reliable and informative measurements of SAM/SAH ratio in live cells. Indeed, while we attempted measurements of SAM in our in vitro enzymatic assay, these were unfortunately inconclusive, which could have several reasons. SAM serves as a crucial component in a multitude of cellular processes, beyond its role in regulating chemotaxis (Loenen, 2006). Therefore, it would be possible that even though DMSP is converted into SAM (via methionine), the SAM pool does not change in a detectable manner because of other concurrent processes that affect SAM levels.

Figure S10. Production of methionine by *Pseudoalteromonas sp. ASV16* cell lysate in an in vitro enzyme assay. Methionine peak intensity measured using mass spectrometry ($[Methionine + H]^+$ m/z 150.0582) for the cell lysate when untreated (a), supplemented with 200 μ M of DMSP (b), 200 μ M of homocysteine (c) and 200 μ M of both DMSP and homocysteine (d). Each treatment was sampled continuously in triplicate ($n = 3$), with line plots representing mean (thick line) \pm SD (shaded region).

1.5. Is DMSP a chemoattractant? Within the chemotactic array there are cooperative interactions between neighboring receptors that recognize different ligands.

We thank the Reviewer for this insightful question. We have addressed this question by testing directly the chemotactic response of our four marine strains to the same concentration range of DMSP used in our laboratory DMSP-addition experiments (i.e., 0.1-10 μ M; Fig. S13, included below for convenience).

*These additional experiments revealed a weak but statistically significant attraction in two of the four strains tested: *Alteromonas* sp. ASV109 and *Pseudoalteromonas* sp. ASV39 (Fig. S13a,b, ANOVA, $p < 0.05$; Table S29). The largest chemotactic response recorded was for *Alteromonas* sp. ASV109 exposed to 0.1 μ M DMSP, with a chemotactic index of 2.45 ± 0.42 (Fig. S13a, ANOVA, $p < 0.05$; Table S29). *Pseudoalteromonas* sp. ASV16 and *Alteromonas* sp. ASV76 did not exhibit chemotaxis to any concentrations of DMSP (Fig. S13c,d, ANOVA, $p > 0.05$; Table S29).*

Although two of our four strains were attracted to DMSP, it is important to highlight that DMSP in our laboratory experiments was only present in the surrounding seawater and was absent from the wells of the ISCA. This means that there were no cooperative interactions between these two ligands.

Figure S13. Chemotactic response of the four marine strains to DMSP_d at different concentrations. Chemotactic index (I_c , coloured bars) of *Alteromonas* sp. ASV109 (a), *Pseudoalteromonas* sp. ASV39 (b), *Pseudoalteromonas* sp. ASV16 (c) and *Alteromonas* sp. ASV76 (d), in response to 0.1, 1 and 10 μ M of DMSP_d in laboratory ISCA experiments. An asterisk denotes positive chemotaxis, i.e. a chemotactic index significantly larger than 1 (ANOVA, $p < 0.05$; Table S29). Each treatment was replicated across three different ISCA (n = 3). Bar plots represent the mean (coloured bar) \pm SD (error bar), with replicates shown as individual dots.

To clarify this point, we have added the following text in Supplementary Note 2:

“DMSP_d in our laboratory experiments was only present in the surrounding seawater and was absent from the wells of the ISCA, and therefore could not induce chemotaxis into the ISCA wells. Nonetheless, we performed additional assays to test the ability of DMSP_d to induce chemotaxis in our four bacterial isolates, at the same concentration range used in our laboratory experiments (0.1-10 μ M; Fig. S13). Among the four strains, we measured a chemotactic index of 2.45 ± 0.42 in *Alteromonas* sp. ASV109 to 0.1 μ M DMSP_d (Fig. S13a, ANOVA, $p < 0.05$; Table S29) and 1.77 ± 0.09 in *Pseudoalteromonas* sp. ASV39 to 1 μ M DMSP_d (Fig. S13b, ANOVA, $p < 0.05$; Table

S29). *Pseudoalteromonas* sp. ASV16 and *Alteromonas* sp. ASV76 did not exhibit chemotaxis to any concentrations of DMSP_d (Fig. S13c,d, ANOVA, $p > 0.05$, Table S29). We highlight again that, although two of the strains tested were weakly attracted to DMSP, this compound was homogenously mixed in the surrounding seawater in our experiments measuring chemotaxis towards laminarin (Fig. 3c-f). Therefore, it could not attract cells into the ISCA wells.“

We have also added further explanation regarding this experiment in the Methods section, under the title of “Chemotaxis assay to DMSP_d”, which reads:

“In order to test whether DMSP_d itself induced chemoattraction in our four marine strains, ISCA experiments were conducted with the same range of concentrations of DMSP_d used in the DMSP_d-addition laboratory experiments (0.1-10 μM ; Fig. S13). A stock solution (10 mM) of DMSP_d was prepared in autoclaved artificial seawater (Instant Ocean, Spectrum Brands) and diluted in individual 15 mL tubes (Falcon) containing 10 mL of the same artificial seawater to final concentrations of i) 0.1 μM , ii) 1 μM and iii) 10 μM . Cell counts of each overnight culture of the four environmental isolates were determined by flow cytometry and cell cultures were diluted in each flask to obtain a cell suspension of 10^6 cells mL^{-1} . For each isolate, three ISCA were each loaded with the three concentrations of DMSP_d, with one row of wells containing artificial seawater as a negative control. The ISCA were incubated for 1 h in the diluted bacterial cultures. Thereafter, the contents of the wells were retrieved and the cells counted by flow cytometry after SYBR Green I staining (ThermoFisher). The chemotactic index was determined from the cell counts as described in “Sample processing - Flow cytometry”.”

In closing, we thank the Reviewer for the insightful comments, and in particular for prodding us to explore in more depth the mechanistic role of DMSP, which has been a gratifying addition to this project and, we hope, a significant addition to the story.

References

- Adler, J. A. Method for measuring chemotaxis and use of the method to determine optimum conditions for chemotaxis by *Escherichia coli*. *Microbiology* **74**, 77–91 (1973).
- Armstrong, J. B. Chemotaxis and methionine metabolism in *Escherichia coli*. *Can. J. Microbiol.* **18**, 591–596 (1972).
- Becker, S. *et al.* Laminarin is a major molecule in the marine carbon cycle. *PNAS.* **117**, 6599–6607 (2020).

- Cantoni, G. L. Methylation of nicotinamide with a soluble enzyme system from rat liver. *J. Biol. Chem.* **189**, 203–216 (1951).
- Caruana, A. M. N. & Malin, G. The variability in DMSP content and DMSP lyase activity in marine dinoflagellates. *Prog. Oceanogr.* **120**, 410–424 (2014).
- Fuhrer, T., Heer, D., Begemann, B. & Zamboni, N. High-throughput, accurate mass metabolome profiling of cellular extracts by flow injection-time-of-flight mass spectrometry. *Anal. Chem.* **83**, 7074–7080 (2011).
- Konishi, H., Hio, M., Kobayashi, M., Takase, R. & Hashimoto, W. Bacterial chemotaxis towards polysaccharide pectin by pectin-binding protein. *Sci. Rep.* **10**, 1–12 (2020).
- Lambert, B. S. *et al.* A microfluidics-based in situ chemotaxis assay to study the behaviour of aquatic microbial communities. *Nat. Microbiol.* **2**, 1344–1349 (2017).
- Loenen, W. A. M. S-Adenosylmethionine: Jack of all trades and master of everything? *Biochem. Soc. Trans.* **34**, 330–333 (2006).
- Lu, S. C. S-Adenosylmethionine. *Int. J. Biochem. Cell Biol.* **32**, 391–395 (2000).
- Mystkowska, A. A. *et al.* Molecular recognition of the beta-glucans laminarin and pustulan by a SusD-like glycan-binding protein of a marine *Bacteroidetes*. *FEBS J.* **285**, 4465–4481 (2018).
- Michel, G., Tonon, T., Scornet, D., Cock, J. M. & Kloareg, B. Central and storage carbon metabolism of the brown alga *Ectocarpus siliculosus*: insights into the origin and evolution of storage carbohydrates in Eukaryotes. *New Phytol.* **188**, 67–81 (2010).
- Miller, T. R., Hnilicka, K., Dziedzic, A., Desplats, P. & Belas, R. Chemotaxis of *Silicibacter* sp. strain TM1040 toward dinoflagellate products. *Appl. Environ. Microbiol.* **70**, 4692–4701 (2004).
- Pontrelli, S. & Sauer, U. Salt-tolerant metabolomics for exometabolomic measurements of marine bacterial isolates. *Anal. Chem.* **93**, 7164–7171 (2021).
- Reintjes, G., Arnosti, C., Fuchs, B. M. & Amann, R. An alternative polysaccharide uptake mechanism of marine bacteria. *ISME J.* **11**, 1640–1650 (2017).
- Reisch, C. R. *et al.* Novel pathway for assimilation of dimethylsulphoniopropionate widespread in marine bacteria. *Nature* **473**, 208–211 (2011).
- Rinaudo, M. 2.21 - Seaweed Polysaccharides. in *Comprehensive Glycoscience* (ed. Kamerling, H.) 691–735 (Elsevier, 2007).
- Stocker, R. & Seymour, J. R. Ecology and Physics of Bacterial Chemotaxis in the Ocean. *Microbiol. Mol. Biol. Rev.* **76**, 792–812 (2012).
- Unfried, F. *et al.* Adaptive mechanisms that provide competitive advantages to marine *Bacteroidetes* during microalgal blooms. *ISME J.* **12**, 2894–2906 (2018).

Response to Reviewer 2

For ease of editorial review, we have included the Reviewers' comments below in black font. Our responses are in blue and *italics*. Text that has been inserted or changed within the modified manuscript is underlined.

This study reports chemotaxis to polysaccharide laminarin and alginate by marine bacteria and enhancement of their chemotaxis by DMSP by serving as methyl donor to MCPs. Finding of chemotaxis to high molecular weight compounds itself is very interesting because chemotaxis towards compounds with low molecular weights have been mainly investigated so far. Actually, findings in this paper will contribute to understanding the carbon cycle in marine environments.

We thank the Reviewer for acknowledging the importance of our work and for their valuable feedback.

Specific comments

2.1. Are polysaccharides themselves chemotactic ligands? It is possible that metabolites derived from laminarin are chemotactic ligands. It is interesting to quantify responses toward culture supernatant on medium containing laminarin.

We thank the Reviewer for their question. As demonstrated in Fig. 2a-d, three out of the four strains we used in laboratory experiments (i.e., ASV16, ASV39 and ASV76) showed the strongest chemotactic response to the largest polymers of laminarin (Fig. 2, ANOVA, $p < 0.05$ in each case; Table S12) and no response to laminarin hexose, laminaribiose or glucose. Similarly, all strains responded significantly more strongly to alginate polymers than alginate oligomers (Fig. S3, ANOVA, $p < 0.05$; Tables S20-S21), while no chemotaxis was found to the monomer mannuronate (except for ASV16, Fig. S3i, ANOVA, $p < 0.05$; Table S21). Our experiments therefore demonstrate the ability of polysaccharides to act as chemotactic ligands for marine bacteria, and a clear preference in the four strains of bacteria tested for these larger molecules compared to their breakdown products.

Quantifying chemotactic responses to culture supernatant containing laminarin would provide results that would likely prove very difficult to untangle. Cell culture media contain a myriad of other compounds, and in this case the supernatant would contain a varying mix of different concentrations of the breakdown products. Any observation of chemotaxis could therefore be in response to other compounds present in the extracts

or supernatant. For this reason, we decided to focus our efforts on different experiments as part of this revision, which we hope the Reviewer will agree was the right choice.

2.2. When DMSP stimulates chemotaxis to polysaccharides by serving as methyl donor to MCPs in marine bacteria, it also stimulates chemotaxis toward chemicals with low molecular weights (like chemotaxis to aspartate in *E. coli*). Does DMSP stimulate chemotaxis of marine bacteria to chemicals with low molecular weights (for example, amino acids or casamino acids)? I consider the possibility that DMSP enhances uptake of polysaccharides, leading to enhancement of chemotaxis to polysaccharides (if it is correct, DMSP enhances growth of marine bacteria on laminarin).

We thank the Reviewer for this question. The chemotactic responses obtained for our four marine strains to the smaller laminarin fraction (<3 kDa), alginate oligomers, glucose and mannuronate provide insights (Figs. 3e-l and S6e-l). Indeed, the chemotactic response of most strains towards these molecules was not significantly affected by the presence of DMSP (ANOVA, $p > 0.05$; Figs. S3e-l and S6e-l, Tables S19-S21). Yet, as the lack of effect of DMSP on chemotaxis towards monomers might be due to the lack of chemotaxis of the strains to the monomers even without DMSP, we followed the Reviewer's input and tested additional small molecules (the amino acids serine and aspartate) with our four marine isolates in the presence of DMSP at DMSP concentrations of 0.1, 1 and 10 μM (Fig. S14a-h, also shown below for convenience). For none of the four strains and none of the three DMSP concentrations the attraction to these two amino acids increased in the presence of DMSP (Fig. S14a-h, ANOVA, $p > 0.05$; Table S31). These results, taken together, suggest that DMSP does not stimulate chemoattraction across all molecules, although we cannot rule out that this may occur for some other molecules that we have not tested.

Figure S14: Chemotactic index (I_c, coloured bars) in response to 1 mM aspartate (a,c,e,g) and 1 mM serine (b,d,f,h) in ISCA laboratory experiments, in the presence and absence of DMSP_d at different concentrations. Experiments were performed without DMSP_d (grey bars) and in the presence of three different concentrations of DMSP_d (0, 0.1, 1 and 10 μM; cyan bars) in the surrounding artificial seawater, for *Pseudoalteromonas* sp. ASV16 (a-b), *Alteromonas* sp. ASV39 (c-d), *Pseudoalteromonas* sp. ASV76 (e-f) and *Alteromonas* sp. ASV109 (g-h). In no case was the chemotactic response significantly larger in the presence of DMSP_d compared to the absence of DMSP_d (ANOVA, $p > 0.05$, Table S31). An asterisk denotes positive chemotaxis, i.e., a chemotactic index significantly larger than 1 (ANOVA, $p < 0.05$; Table S31). Each treatment was replicated across three different ISCA (n = 3). Bar plots represent the mean (colored bar) ± SD (error bar), with replicates shown as individual dots.

We also tested another suite of compounds in the presence of DMSP, including spermidine, trimethylamine (TMA) and 10% Marine Broth 2216 (Fig. S15a-c), a rich medium containing a mixture of Peptone and yeast extract. These compounds have been identified as strong chemoattractants in marine bacteria (Clerc, Schreier et al., in prep.). While all three compounds induced a significant chemotactic response (Fig. S15a-c, ANOVA, $p < 0.05$; Table S31), only in one case (spermidine) did DMSP (1 μM) increase chemotaxis, compared to the case without DMSP (see the bar in a darker shade, Fig. S15b, ANOVA, $p < 0.05$; Table S31). While limited of course to the set of compounds we tested, these data suggest that the enhancement of chemotaxis by DMSP may apply primarily or be strongest in polysaccharides. Ecologically, this would be relevant in helping bacteria climb the steeper gradients expected for larger molecules (compared to smaller molecules) in view of their lower diffusivity.

Figure S15. Chemotactic response of *Pseudoalteromonas sp. ASV16* to 10% Marine Broth and two small compounds in the presence of different concentrations of DMSP_d. Chemotactic index (I_c, coloured bars) of *Pseudoalteromonas sp. ASV16* in response to 10% Marine Broth 2216 (a), spermidine (b) and trimethylamine (TMA, c), in the presence of different concentrations of DMSP (0, 0.1, 1 and 10 μM) in the surrounding artificial seawater. An asterisk denotes a chemotactic index significantly larger than 1 (ANOVA, $p < 0.05$; Table S31). The bar in darker shade indicates that the chemotactic response to spermidine was significantly larger than the treatment without DMSP_d (ANOVA, $p < 0.05$; Table S31). Each treatment was replicated across three different ISCA's ($n = 3$). Bar plots represent the mean (coloured bar) \pm SD (error bar), with replicates shown as individual dots. To clarify this point, we have added the results of these experiments as Supplementary Note 4, which reads:

“In order to further elucidate whether the impact of DMSP on chemotaxis is limited to large polymers, we tested two additional small molecules -- the amino acids serine and aspartate -- with our four marine isolates in the presence of DMSP at different concentrations (Fig. S14a-h). No chemotactic response towards these two amino acids was measured and, additionally, no difference in attraction was recorded in the presence of DMSP (Fig. S14, ANOVA, $p > 0.05$; Table S31). These results suggest that DMSP does not stimulate chemoattraction towards molecules that cells are not already attracted to. Further experiments to test the generality of the enhancing effect of DMSP were conducted using two other small compounds, spermidine and trimethylamine (TMA), as well as 10% Marine Broth 2216 (Fig. S15a), a rich medium containing a mixture of Peptone and yeast extract. While all three compounds induced a significant chemotactic response (Fig. S15a-c, ANOVA, $p < 0.05$; Table S31), only spermidine induced a significantly greater response in the presence of 1 μM DMSP compared to the case without DMSP (Fig. S15b, ANOVA, $p < 0.05$; Table S31). While limited to the set of compounds we tested, these data suggest that the enhancement of chemotaxis by DMSP may apply primarily or be strongest for polysaccharides. Ecologically, this would be relevant in helping bacteria climb the steeper gradients expected for larger molecules in view of their lower diffusivity.”

We have also modified the Methods section “Chemotaxis experiments in the presence of DMSP, methionine, choline or homocysteine in the surrounding seawater” to describe these additional experiments. It now reads:

“Additional tests in the presence of DMSP were run using aspartate (1 mM, Sigma-Aldrich), serine (1 mM, Sigma-Aldrich), spermidine (1 mM, Sigma-Aldrich), trimethylamine (TMA, 1mM, Sigma-Aldrich) and 10% Marine Broth 2216 (Sigma-Aldrich).”

In closing, we thank the Reviewer for the insightful comments. We have in particular found it helpful to contrast the enhancing effects of DMSP on polymers versus smaller molecules, which we believe adds an interesting dimension to the story.

References

Clerc, E. E., Schreier J. E., Slomka, J., Smith, C. B., Fu, H., Seymour, J. R., Raina, J.-B., Moran M. A., & Stocker R. Signals and substrates: The role of chemotaxis and growth in shaping marine bacterial community composition. In prep.

Response to Reviewer 3

For ease of editorial review, we have included the Reviewers' comments below in black font. Our responses are in blue and *italics*. Text that has been inserted or changed within the modified manuscript is underlined.

Clerc *et al.* demonstrated that laminarin can elicit strong chemoattraction, which can be further enhanced by the important organic sulfur compound DMSP, by their in situ chemotaxis assay (ISCA). This work for the first time indicated a potential role of DMSP in promoting bacterial chemoattraction, which could be an important finding. The authors also provide an explanation of the possible mechanism behind this role of DMSP. The manuscript is generally well-written. However, in my opinion, some questions remained to be answered in this manuscript regarding the DMSP related results, which might make their conclusions more reliable.

We thank the Reviewer for their insightful feedback and their acknowledgement of the importance of our findings. We have performed additional experiments to reinforce our findings concerning the influence of DMSP on bacterial chemotaxis and have revised the manuscript to reflect this additional evidence.

Major points

3.1. DMSP could promote chemoattraction is the most important conclusion of this paper. However, it seems that the authors do not fully describe the importance of DMSP in their introduction, especially about its role as a chemical cue in previous publications. This information has only been mentioned briefly in the first paragraph and later in the results and discussion, which should be better to expand this somewhere appropriate. I think this information is necessary for the readers to understand the importance of this paper.

We thank the Reviewer for this suggestion – we may indeed have taken the importance of DMSP too much for granted, whereas it is definitely useful to bring this to the attention in particular of a wider audience. We have now amended the main text to provide more background on the ecological importance of DMSP (while exerting some restraint on the length of the additional text). The relevant section now reads as follows:

*“This observation was intriguing given that DMSP is itself a potent behavioural cue (Seymour *et al.*, 2010; Garren *et al.*, 2014). This molecule is widely produced by*

phytoplankton, is one of the most abundant reduced sulfur compounds in the ocean (Malin et al., 1993; Kwint and Kramer, 1996) and an important nutrient source for marine microorganisms (Kiene et al., 2000). In addition, DMSP has often been reported for its chemoattractive properties, not only for bacteria, but also for marine protists (Seymour et al., 2010; Garren et al., 2014; Miller et al., 2004) and even fishes (DeBose et al., 2008). The multifaceted ecological importance of DMSP has been described in several reviews (Reisch et al., 2011; Curson et al., 2011).”

3.2. Line 173-183: Can you rule out the possibility that the observed higher I_C of laminarin is affected by the effect related to the stimulated growth? Moreover, when you discussed about the chemoattraction promotion effect of DMSP (in line 250-252), have you ruled out the possibility that DMSP may promote the bacterial growth since it is an important carbon source?

Yes, we believe we can confidently rule out this possibility, and we have performed additional experiments to convey this point most clearly in the manuscript. Our ISCA experiments last only one hour in total. In a set of new experiments, we calculated the doubling time of each of the four isolates grown on laminarin as the sole carbon source, in the same culture conditions to an ISCA experiment, and found doubling times of 14.4 ± 1.6 hours, 17.3 ± 0.4 hours and 33.7 ± 3.3 hours for ASV109, ASV16 and ASV76, respectively (Fig. S2, Table S14; no growth was measured for ASV39, reflecting results already presented in Fig. 2f). We therefore conclude that the number of cells in the ISCA wells after 1 hour of incubation results from chemotaxis and not growth in the ISCA.

Figure S2. Growth curves for ASV16 (green), ASV39 (blue), ASV76 (purple) and ASV109 (red) on 10 mg mL⁻¹ laminarin, measured using a plate reader over 48 h. From an overnight rich culture, cells were diluted 1:100 in artificial seawater containing 10 mg mL⁻¹ laminarin, the highest concentration used in ISCA wells in our experiments. Experiments were performed in triplicate: thick solid lines represent the mean and shaded regions denote the standard deviation.

We have added the following information to the main text:

“Over all four strains, growth rates on laminarin were on average 3.8 times higher than on laminaribiose and 3.0 times higher than on glucose (Fig. 2e-h, Table S13). Yet importantly, growth on laminarin within the timescale of an ISCA experiment (1 h) was negligible for all four strains (Fig. S2, Table S14), demonstrating that the number of cells in ISCA wells is a result of chemotaxis and not growth.”

Furthermore, regarding growth on DMSP, it is important to highlight that in our experiments DMSP was added to the surrounding seawater, not inside the ISCA wells. If DMSP stimulated growth within the timescale of the experiments (1 h), we would expect an increase in the cell count in the artificial seawater control of the ISCA upon

DMSP amendment. This was not the case (Fig. 3c-f, ANOVA, $p > 0.05$ for all strains, Table S30).

We further confirmed that cells do not appreciably grow on DMSP over the course of 1 h, by incubating the four marine strains in artificial seawater amended with 10 μ M DMSP, the highest concentration used in our experiments, and measuring growth with a plate reader for 48 h (Fig. S5, Table S18). The strains exhibited doubling times of 172.0 ± 36.1 hours, 192 ± 23.8 hours, 502.0 ± 72.4 hours and 928.9 ± 402.3 hours for ASV109, ASV16, ASV76 and ASV39, respectively (Fig. S5, Table S18), indicating very little growth on DMSP occurred in our experiments.

Figure S5. Growth curves for ASV16 (green), ASV39 (blue), ASV76 (purple) and ASV109 (red) on 10 mM DMSP, measured using a plate reader over 48 h. From an overnight rich culture, cells were diluted 1:100 in artificial seawater containing 10 mM DMSP to mimic the conditions outside ISCA wells in some of our experiments. Experiments were performed in triplicate: thick solid lines represent the mean and shaded regions denote the standard deviation.

To clarify that the concentrations of DMSP used in our experiments were too low to induce growth within 1 h, we added the following to the main text:

“We quantified their chemotactic response towards laminarin (10 mg mL⁻¹) when the background artificial seawater (outside of the ISCA wells, Supplementary Note 2) was spiked with DMSP_d at different concentrations (0.1, 1 and 10 μM), representative of phytoplankton lysis events (Caruana and Malin, 2014), yet not sufficient to induce any appreciable bacterial growth within 1 h (Fig. S5, Table S18).

We have also added a description of the growth assays, both on laminarin and on DMSP, in the Methods section under “Growth assays”:

“To confirm that the cell number in the ISCA wells upon amendment of DMSP or laminarin was not due to growth of the bacterial isolates, we ran growth assays replicating how cells are prepared in ISCA experiments. Overnight cultures of the four environmental isolates were grown in 10% Difco 2216 Marine Broth medium (BD Diagnostics) and were subsequently diluted 1/100 in 10 mL tubes containing artificial seawater amended with i) 10 mg mL⁻¹ laminarin, ii) 10 μM DMSP or iii) no amendment as negative control. All treatments were conducted in triplicate. Bacterial growth was measured at 27 °C in a shaking plate reader for 48 h.”

3.3. Many marine bacteria are involved in DMSP degradation and producing DMS or MeSH. Have the authors confirmed that their *Pseudoalteromonas* and *Alteromonas* strains were able to catabolize DMSP or not? Otherwise there might also be the influence of resulting DMS or MeSH? As far as we know at least some of isolates in these two genera have such capacity.

We thank the Reviewer for this question. Genes involved in DMSP catabolism, such as the dmdABCD gene operon and the ddd genes (dddD, dddK, dddL, dddP, dddQ, dddY, and dddW) have mostly been identified in marine members of the class Alphaproteobacteria, namely in the Roseobacter group, the SAR11 clade and the SAR116 cluster (Reisch et al., 2011; Howard et al., 2006; Moran et al., 2012; Sun et al., 2016; Oh et al., 2010). As the genomes of our four marine isolates have not been sequenced, we determined whether these genes were present in the genomes of closely related strains.

We first ran a sequence similarity BLAST search on the KEGG database, querying the 16S rRNA gene sequences of our four isolates, to identify the genomes of the closest sequenced bacteria (minimum similarity: 98.02%, E-value for all: 0.0; Table S26). We then used ratified Gammaproteobacterial sequences for each DMSP catabolism gene from the KEGG database to investigate their presence in the genome of the closest

sequenced relative of each of our isolates (minimum E-value: 3e-05; Table S26). Finally, we performed a reciprocal best hits (RBH) BLAST analysis of the identified gene sequences in the genome from which the ratified DMSP catabolism genes were derived (Table S26) to verify the robustness of the orthology of the top hits.

Our analysis revealed that the only gene involved in DMSP catabolism that was a reciprocal best hit was *dmdC* (in the closest relative of *Alteromonas* ASV109 and ASV76 only; Table S26). *DmdC* is part of the demethylation pathway but mediates a reaction downstream of DMSP degradation (converting 3-methylmercaptopropionyl-CoA into methylthioacryloyl-CoA). No other genes involved in DMSP catabolism were identified through our RBH analysis (Table S26). These results therefore suggest that the strains used in our experiments do not possess the commonly known demethylation or cleavage pathways involved in DMSP catabolism, yet are able to metabolize DMSP. Using an *in vitro* enzyme assay, we further validated this hypothesis and demonstrated that *Pseudoalteromonas* sp. ASV16 is able to catabolize DMSP into methionine, likely through a yet uncharacterized pathway (Fig. S10). These results are fully described in response to Question 3.5 below.

In order to clarify the point made by the Reviewer, we modified the main text as follows:

“This reaction is likely mediated through a yet uncharacterized pathway, as all close relatives of the marine strains used here lack the *dmdA* gene, the only gene known to be capable of demethylating DMSP (Reisch et al. 2011; Table S26).”

Furthermore, we added our gene orthology search in the Methods section under “Orthology of DMSP catabolism genes”:

“To determine whether genes involved in DMSP catabolism are present in our bacterial strains, the 16S rRNA gene sequences of the four isolates were used to identify genomes of the closest sequenced organisms using BLASTn on the KEGG database (minimum similarity: 98.02%; Kanehisa and Goto, 2000). Each DMSP catabolism gene (*dmdA*, *dmdB*, *dmdC*, *dmdD*, *dddD*, *dddP*, *dddY*, *dddQ*, *dddW* and *dddL*) was then queried in the KEGG database (Kanehisa and Goto, 2000) and the sequences harboured by Gammaproteobacterial genomes were selected. Each DMSP-degrading gene was used in a BLAST_p analysis (KEGG, Kanehisa and Goto, 2000) to search for orthologous sequences in the closest relatives of our marine isolates (Table S26). Finally, a reciprocal best hits BLAST of the identified orthologous sequences was carried out in the genomes of the Gammaproteobacteria from which these DMSP-degrading genes originated (Table S26).”

3.4. DMSP itself has been reported to stimulate chemotactic responses. Have you tested that the chemoattraction effect of DMSP in your ISCA?

We thank the reviewer for this question. To address this question, we directly examined the chemotactic response of our four marine strains to DMSP. We utilized the same concentration range as in our DMSP-addition laboratory experiments (i.e., 0.1-10 μ M, Fig. S13, included below for convenience).

*These additional experiments revealed a weak but statistically significant attraction in two of the four strains tested: *Alteromonas* sp. ASV109 and *Pseudoalteromonas* sp. ASV39 (Fig. S13, ANOVA, $p < 0.05$; Table S29). The largest chemotactic response recorded was for *Alteromonas* sp. ASV109 exposed to 0.1 μ M DMSP, with a chemotactic index of 2.45 ± 0.42 (Fig. S13a, ANOVA, $p < 0.05$; Table S29). *Pseudoalteromonas* sp. ASV16 and *Alteromonas* sp. ASV76 did not exhibit chemotaxis to any concentrations of DMSP (Fig. S13c,d, ANOVA, $p > 0.05$; Table S29).*

Although two of our four strains were attracted to DMSP, it is important to highlight that DMSP in our laboratory experiments was only present in the surrounding seawater and was absent from the wells of the ISCA. This means that there were no cooperative interactions between these two ligands.

Figure S13. Chemotactic response of the four marine strains to DMSP_d at different concentrations. Chemotactic index (I_c , coloured bars) of *Alteromonas* sp. ASV109 (a), *Pseudoalteromonas* sp. ASV39 (b), *Pseudoalteromonas* sp. ASV16 (c) and *Alteromonas* sp. ASV76 (d), in response to 0.1, 1 and 10 μM of DMSP_d in laboratory ISCA experiments. An asterisk denotes positive chemotaxis, i.e. a chemotactic index significantly larger than 1 (ANOVA, $p < 0.05$; Table S29). Each treatment was replicated across three different ISCAs ($n = 3$). Bar plots represent the mean (colored bar) \pm SD (error bar), with replicates shown as individual dots.

To clarify the point of the Reviewer, we have added the following in Supplementary Note 2:

“DMSP_d in our laboratory experiments was only present in the surrounding seawater and was absent from the wells of the ISCA, and therefore could not induce chemotaxis into the ISCA wells. Nonetheless, we performed additional assays to test the ability of DMSP_d to induce chemotaxis in our four bacterial isolates, at the same concentration range used in our laboratory experiments (0.1-10 μM ; Fig. S13). Among the four strains, we measured a chemotactic index of 2.45 ± 0.42 in *Alteromonas* sp. ASV109 to 0.1 μM DMSP_d (Fig. S13a, ANOVA, $p < 0.05$; Table S29) and 1.77 ± 0.09 in

Pseudoalteromonas sp. ASV39 to 1 μ M DMSP_d (Fig. S13b, ANOVA, $p < 0.05$; Table S29). *Pseudoalteromonas* sp. ASV16 and *Alteromonas* sp. ASV76 did not exhibit chemotaxis to any concentrations of DMSP_d (Fig. S13c,d, ANOVA, $p > 0.05$, Table S29). We highlight again that, although two of the strains tested were weakly attracted to DMSP, this compound was homogenously mixed in the surrounding seawater in our experiments measuring chemotaxis towards laminarin (Fig. 3c-f). Therefore, it could not attract cells into the ISCA wells.

We have also added further explanation regarding this experiment in the Methods section, under the title of “Chemotaxis assay to DMSP_d”, which reads:

“In order to test whether DMSP_d itself induced chemoattraction in our four marine strains, ISCA experiments were conducted with the same range of concentrations of DMSP_d used in the DMSP_d-addition laboratory experiments (0.1-10 μ M; Fig. S13). A stock solution (10 mM) of DMSP_d was prepared in autoclaved artificial seawater (Instant Ocean, Spectrum Brands) and diluted in individual 15 mL tubes (Falcon) containing 10 mL of the same artificial seawater to final concentrations of i) 0.1 μ M, ii) 1 μ M and iii) 10 μ M. Cell counts of each overnight culture of the four environmental isolates were determined by flow cytometry and cells were diluted in each flask to obtain a cell suspension of 10^6 cells mL⁻¹. For each isolate, three ISCA wells were each loaded with the three concentrations of DMSP_d, with one row of wells containing artificial seawater as a negative control. The ISCA wells were incubated for 1 h in the diluted bacterial cultures. Thereafter, the contents of the wells were retrieved and the cells counted by flow cytometry after SYBR Green I staining (ThermoFisher). The chemotactic index was determined from the cell counts as described in “Sample processing - Flow cytometry”.”

3.5. The authors assumed that DMSP acts as a source of methyl groups. However, it seems that there are no publications about methyltransferases specific on DMSP. It is not like methionine, whose methyltransferases were widely existed in diverse bacteria. Although the authors provided indirect evidence based on the experiment with other compounds such as choline, it would be better if the authors can provide more direct evidence. For example, how could methyl transfer happen with DMSP or what is the resulting product?

*We thank the Reviewer for pushing us to provide more evidence to support our claims. In the manuscript, we postulate that DMSP serves as a source for methylation of the chemotaxis receptors, facilitated through the methionine-SAM pathway. The Reviewer points out correctly that no pathway directly transforming DMSP into methionine has been characterized, except through the *dmdABCD* genes (Reisch et al., 2011). Given the results of BLAST analysis of these genes on closely related genomes (Table S26), it*

is unlikely that our strains are able to convert DMSP into methionine through this pathway.

*To test the ability of our bacterial isolates to catabolize DMSP, we conducted in vitro enzyme assays (Methods) employing *Pseudoalteromonas* sp. ASV16 cell lysate as the enzymatic source. This experimental approach allowed us to delve into the enzymatic activity associated with the conversion process.*

In these assays, we monitored the production of methionine using MS after exposing the cell lysate to different molecules. We observed an increase in methionine levels only when both homocysteine and DMSP were added to the extract in combination, and not for either molecule alone (Fig. S10d). These results provide direct evidence that ASV16 contains enzymes that perform methyl transfer from DMSP to homocysteine to form methionine. Methionine can then be converted to SAM (Cantoni, 1951; Armstrong, 1972; Lu, 2000), which binds to CheR and enhances chemotaxis (Adler, 1973).

Figure S10. Production of methionine by *Pseudoalteromonas sp. ASV16* cell lysate in an in vitro enzyme assay. Methionine peak intensity measured using mass spectrometry ($[Methionine + H]^+$ m/z 150.0582) for the cell lysate when untreated (a), supplemented with 200 μ M DMSP (b), 200 μ M homocysteine (c) and 200 μ M of both DMSP and homocysteine (d). Each treatment was sampled continuously in triplicate ($n = 3$), with line plots representing mean (thick line) \pm SD (shaded region).

We have modified the main text to include these results, as follows:

“Finally, using an in vitro enzyme assay employing *Pseudoalteromonas sp. ASV16* cell lysate as an enzymatic source, we demonstrated that the cell lysate is capable of producing methionine only when supplied with DMSP and homocysteine in combination

(Fig. S10d). These results provide direct evidence that ASV16 possesses enzymes able to transfer a methyl group from DMSP to homocysteine to form methionine. This reaction is likely mediated through a yet uncharacterized pathway, as all close relatives of the marine strains used here lack the dmdA gene, the only gene known to be capable of demethylating DMSP (Reisch et al. 2011; Table S26)."

In order to clarify that the conversion of methionine into SAM is a common metabolic process in bacteria, we have also added the following statement:

"These studies revealed that the sensitivity of chemoreceptors (methyl-accepting chemotaxis proteins, MCPs) in *E. coli* strongly depends on the availability of methyl groups provided by S-adenosyl-methionine (SAM), a methyl donor that is derived from methionine (Armstrong, 1972), which occurs in high intracellular concentration in bacteria and is involved in many cellular processes (Cantoni, 1951; Armstrong, 1972; Lu, 2000)."

Further, we have added the enzyme assay in the Methods section, which now reads:

"In order to identify the mechanism by which the addition of DMSP affects chemotaxis, we performed an in vitro enzyme assay to determine which substrates induce the production of methionine by *Pseudoalteromonas sp. ASV16* cell lysate. *Pseudoalteromonas sp. ASV16* cells were grown overnight in rich medium (10% Marine Broth 2216; Sigma-Aldrich) and resuspended in artificial seawater (Instant Ocean, Spectrum Brands) at 10^6 cells mL^{-1} in a 50 mL tube (Falcon). Three replicate experiments were performed. DMSP (10 μM , Sigma-Aldrich) and laminarin (10 mg mL^{-1} , Sigma-Aldrich) were added to the cell cultures, which were then incubated for 1 h at room temperature, to mimic the conditions of an ISCA experiment. After this incubation period, the cell cultures were spun down (3000 rpm, 20 min) and resuspended in artificial seawater. This washing step was repeated a second time and the cell pellet was then resuspended in 250 μL of MilliQ water containing Roche Complete protease inhibitor (Sigma-Aldrich) before it was snap-frozen in liquid nitrogen and stored at -80°C until the assay was performed.

Before use, the resuspended cells were thawed at room temperature for 30 min to promote cell lysis, then briefly vortexed and transferred to ice for another 15 min. The cells were then centrifuged at 10,000 rpm for 5 min and the supernatant was collected and used as the protein extract for the in vitro enzyme assay. DMSP methyltransferase activity was tested using an in vitro enzyme assay, using 200 μM of DMSP and homocysteine in four different combinations: DMSP only, homocysteine only, both, and none of the two. Each substrate combination was diluted in 20 mM of ammonium bicarbonate buffer (pH 7.8) in triplicate treatments. The assay was initiated by adding 10 μL of the protein extract to 90 μL of the enzyme assay mix. The enzyme assay was

performed in the autosampler of an Agilent 1290 Infinity LC stack kept at 18°C and formation of methionine was monitored by sampling the enzyme reaction continuously over time.

Measurements were performed using Liquid Chromatography coupled with an Agilent 6546 Quadrupole Time of Flight Mass Spectrometer in positive mode, 10 GHz, high resolution mode. An Agilent EC-CN Poroshell column (50 mm × 2.1 mm, 2.7 μM) was used isocratically to reduce interference of salts on metabolite ionization (Pontrelli and Sauer, 2021). The buffer consisted of 10% acetonitrile (CHROMASOLV) in 90% water with 0.1% formic acid (Sigma-Aldrich), with a flow rate of 1 mL min⁻¹ at 20°C. Every 2 min, a 3 μL sample was injected into the instrument. Raw data was treated with a spectral processing and alignment pipeline using Matlab (The Mathworks, Natick) as described previously (Fuhrer et al., 2011).”

All metabolomics raw spectral files have been deposited into the MassIVE database with the accession code MSV000092825 and password "reviewer123". The dataset will be made publicly accessible upon acceptance.

In closing, we thank the Reviewer for their time and insights. We hope we have addressed them satisfactorily and we believe that they have resulted in an improved manuscript.

References

- Adler, J. A. Method for measuring chemotaxis and use of the method to determine optimum conditions for chemotaxis by *Escherichia coli*. *Microbiology* **74**, 77–91 (1973).
- Armstrong, J. B. Chemotaxis and methionine metabolism in *Escherichia coli*. *Can. J. Microbiol.* **18**, 591–596 (1972).
- Cantoni, G. L. Methylation of nicotinamide with a soluble enzyme system from rate liver. *J. Biol. Chem.* **189**, 203–216 (1951).
- Caruana, A. M. N. & Malin, G. The variability in DMSP content and DMSP lyase activity in marine dinoflagellates. *Prog. Oceanogr.* **120**, 410–424 (2014).
- Curson, A. R. J., Todd, J. D., Sullivan, M. J. & Johnston, A. W. B. Catabolism of dimethylsulphonioacetate: microorganisms, enzymes and genes. *Nature Rev. Microbiol.* **9**, 849–859 (2011).
- DeBose, J. L., Lema, S. C. & Nevitt, G. A. Dimethylsulfonylacetate as a foraging cue for reef fishes. *Science* **319**, 1356 (2008).

- Fuhrer, T., Heer, D., Begemann, B. & Zamboni, N. High-throughput, accurate mass metabolome profiling of cellular extracts by flow injection-time-of-flight mass spectrometry. *Anal. Chem.* **83**, 7074–7080 (2011).
- Garren, M. *et al.* A bacterial pathogen uses dimethylsulfoniopropionate as a cue to target heat-stressed corals. *ISME J.* **8**, 999–1007 (2014).
- Howard, E. C. *et al.* Bacterial taxa that limit sulfur flux from the ocean. *Science* **314**, 649–652 (2006).
- Kanehisa, M. & Goto, S. KEGG: Kyoto Encyclopedia of Genes and Genomes. *Nucleic Acids Res.* **28**, 27–30 (2000).
- Kiene, R. P., Linn, L. J. & Bruton, J. A. New and important roles for DMSP in marine microbial communities. *J. Sea Res.* **43**, 209–224 (2000).
- Kwint, R. L. J. & Kramer, K. Annual cycle of the production and fate of DMS and DMSP in a marine coastal system. *Mar. Ecol. Prog.* **134**, 217–224 (1996).
- Lu, S. C. S-Adenosylmethionine. *Int. J. Biochem. Cell Biol.* **32**, 391–395 (2000).
- Malin, G., Turner, S., Liss, P., Holligan, P. & Harbour, D. Dimethylsulphide and dimethylsulphonioacetate in the Northeast Atlantic during the summer coccolithophore bloom. *Deep Sea Res. Part I. Oceanogr. Res.* **40**, 1487–1508 (1993).
- Miller, T. R., Hnilicka, K., Dziedzic, A., Desplats, P. & Belas, R. Chemotaxis of *Silicibacter* sp. strain TM1040 toward dinoflagellate products. *Appl. Environ. Microbiol.* **70**, 4692–4701 (2004).
- Moran, M. A., Reisch, C. R., Kiene, R. P. & Whitman, W. B. Genomic insights into bacterial DMSP transformations. *Annu. Rev. Mar. Sci.* **4**, 523–542 (2012).
- Oh, H.-M. *et al.* Complete genome sequence of “*Candidatus Puniceispirillum marinum*” IMCC1322, a representative of the SAR116 clade in the Alphaproteobacteria. *J. Bacteriol.* **192**, 3240–3241 (2010).
- Pontrelli, S. & Sauer, U. Salt-tolerant metabolomics for exometabolomic measurements of marine bacterial isolates. *Anal. Chem.* **93**, 7164–7171 (2021).
- Reisch, C. R. *et al.* Novel pathway for assimilation of dimethylsulphonioacetate widespread in marine bacteria. *Nature* **473**, 208–211 (2011).
- Seymour, J. R., Simó, R., Ahmed, T. & Stocker, R. Chemoattraction to dimethylsulfoniopropionate throughout the marine microbial food web. *Science* **329**, 342–345 (2010).

Sun, J. *et al.* The abundant marine bacterium *Pelagibacter* simultaneously catabolizes dimethylsulfoniopropionate to the gases dimethyl sulfide and methanethiol. *Nat. Microbiol.* **1**, 1–5 (2016).

Reviewers' Comments:

Reviewer #1:

Remarks to the Author:

The authors have addressed satisfactorily my concerns. My major concern was the vague statement in the initial version of the manuscript on the molecular mechanism of DMSP "that DMSP enhances chemotaxis towards laminarin by supplying methylgroups to MCPs". The authors have conducted the suggested mass spectrometry experiments and were able to show that DMSP is essential for the synthesis of L-Met from S-adenosyl-homocysteine. These experiments significantly advance our understanding of the molecular mechanism of DMSP.

Reviewer #2:

Remarks to the Author:

This paper provides two important information, chemotaxis to high molecular weight compounds and enhancement of chemotaxis by DMSP. The authors properly respond to my comments and have revised manuscript. I consider this paper is valuable for publication in Nature communications.

Reviewer #3:

Remarks to the Author:

In the revised version of this manuscript, the authors have clearly answered the questions raised by the reviewers. These new evidences make this study more solid and comprehensively. The authors have thoroughly addressed my questions regarding the possibility of laminarin or DMSP to stimulate cell growth, and chemotactic responses to DMSP. More strikingly, the authors find that *Pseudoalteromonas* sp. ASV16 was able to transfer a methyl group from DMSP to homocysteine to form methionine, which further support their claims. It is a pity that the authors did not directly quantify DMSP catabolism in these four strains but just searched the genomes of closely related strains, but this is acceptable. I greatly appreciate the author's rigorous research attitude, and we have gained a lot during the process of reviewing this article.